# Intestinal GCN2 controls *Drosophila* systemic growth in response to *Lactiplantibacillus plantarum* symbiotic cues encoded by r/tRNA operons

Théodore Grenier[1]*[†], Jessika Consuegra[1], Mariana G Ferrarini[2,3], Houssam Akherraz[1], Longwei Bai[1], Yves Dusabyinema[1], Isabelle Rahioui[2], Pedro Da Silva[2], Benjamin Gillet[1], Sandrine Hughes[1], Cathy I Ramos[1], Renata C Matos[1], François Leulier[1]*

[1]Institut de Génomique Fonctionnelle de Lyon, Ecole Normale Superieure de Lyon, Université Claude Bernard, Lyon, France; [2]Univ Lyon, INSA Lyon, INRAE, BF2I, UMR 203, 69621, Villeurbanne, France; [3]Laboratoire de Biométrie et Biologie Évolutive, UMR 5558, Université Lyon 1, Université Lyon, Villeurbanne, France

**\*For correspondence:**
t.grenier@hubrecht.eu (TG);
francois.leulier@ens-lyon.fr (FL)

**Present address:** [†]Hubrecht Institute, Royal Netherlands Academy of Arts and Sciences (KNAW), University Medical Center Utrecht, 3584 CT Utrecht, the Netherlands, Utrecht, Netherlands

**Competing interest:** The authors declare that no competing interests exist.

**Abstract** Symbiotic bacteria interact with their host through symbiotic cues. Here, we took advantage of the mutualism between *Drosophila* and *Lactiplantibacillus plantarum* (Lp) to investigate a novel mechanism of host-symbiont interaction. Using chemically defined diets, we found that association with Lp improves the growth of larvae-fed amino acid-imbalanced diets, even though Lp cannot produce the limiting amino acid. We show that in this context Lp supports its host's growth through a molecular dialogue that requires functional operons encoding ribosomal and transfer RNAs (r/tRNAs) in Lp and the general control nonderepressible 2 (GCN2) kinase in *Drosophila*'s enterocytes. Our data indicate that Lp's r/tRNAs are packaged in extracellular vesicles and activate GCN2 in a subset of larval enterocytes, a mechanism necessary to remodel the intestinal transcriptome and ultimately to support anabolic growth. Based on our findings, we propose a novel beneficial molecular dialogue between host and microbes, which relies on a non-canonical role of GCN2 as a mediator of non-nutritional symbiotic cues encoded by r/tRNA operons.

## Editor's evaluation

Previous studies found that a component of the microbiota, *Lactobacillus plantarum*, can provide support to its host *Drosophila melanogaster* during development. Here, the authors further explore this interaction using defined diets where they find that under conditions that have low levels of some essential amino acids, the bacteria can still promote survival even though the bacteria is not synthesizing the amino acid. Through a screen of bacterial transposon insertion mutants, these authors identify bacterial transfer and ribosomal RNAs as necessary for this effect, and studies in the fly demonstrate that the host kinase GCN2, a protein known to associate with host tRNAs, in enterocytes is the mediator of this response. This manuscript links the intestinal microbiota to host protective responses providing important insights into these interactions.

## Introduction

Animals have evolved and live in symbiosis with a great diversity of microbes. Symbiotic microbes strongly influence various aspects of their host's physiology, metabolism, and behaviour. The fruit

fly *Drosophila melanogaster* (hereinafter referred to as *Drosophila*) is a powerful model to study the mechanisms underlying the interactions between host and symbiotic microbes. Indeed, *Drosophila* harbours simple bacterial communities, which individual components can be cultured aerobically. Moreover, *Drosophila* can easily be bred axenically, allowing gnotobiotic studies. Finally, *Drosophila*'s main bacterial symbionts can be genetically engineered, enabling deep mechanistical studies on both the host side and the microbe side. In the past decade, *Drosophila*'s symbiotic microbes were shown to modulate their host's post-embryonic growth (*Shin et al., 2011*; *Storelli et al., 2011*), reproduction (*Elgart et al., 2016*; *Gould et al., 2018*), lifespan (*Keebaugh et al., 2019*; *Yamada et al., 2015*), metabolism (*Gnainsky et al., 2021*; *Kamareddine et al., 2018*; *Newell and Douglas, 2014*), immunity (*Iatsenko et al., 2018*), social behaviour (*Chen et al., 2019*; *Sharon et al., 2010*), and food preference (*Kim et al., 2021*; *Leitão-Gonçalves et al., 2017*). The mechanisms underpinning these phenomena often rely on symbiotic cues: molecules produced by symbiotic microbes that interact with the host's signalling pathways. Symbiotic cues can be nutrients: for instance, amino acids (AA) produced by symbiotic bacteria can inhibit the production of the neuropeptide CNMamide in the gut, which represses preference for AA (*Kim et al., 2021*). In addition, symbiotic cues can be non-nutritional. Sensing of symbiotic bacteria's cell wall components by gut cells leads to the production of digestive enzymes, which helps the larva digest the dietary polypeptides and improves its systemic growth (*Erkosar et al., 2015*; *Matos et al., 2017*). Acetate produced by symbiotic microbes alters the epigenome of enteroendocrine cells, which stimulates the secretion of the hormone tachykinin (*Jugder et al., 2021*). Tachykinin then promotes lipid utilisation in nearby enterocytes (*Kamareddine et al., 2018*). Beyond these examples, we posit that symbiotic cues may be widespread in *Drosophila*-microbes symbiosis but their nature remains elusive. Hence, identifying them is an important goal of the field of host-symbionts interactions (*Selosse et al., 2014*).

In this study, we sought to identify additional non-nutritional symbiotic cues that allow *Drosophila*'s symbiotic bacteria to influence the physiology of their host. As a readout, we used *Drosophila*'s post-embryonic systemic growth. *Drosophila*'s growth phase (larval stages) depends widely on the nutritional environment: larvae reach metamorphosis (pupariation) in 4–5 days in optimal nutritional conditions or up to 15–20 days in severe malnutrition conditions (*Erkosar et al., 2013*; *Tennessen and Thummel, 2011*). Nutrition regulates larval systemic growth through the action of nutrient-sensing regulatory pathways, especially AA-sensing pathways. The main AA (or lack of) sensing pathways are the target-of-rapamycin (TOR) kinase pathway and the general control nonderepressible 2 (GCN2) kinase (*Gallinetti et al., 2013*). Both kinases were first described in yeast (*Dever et al., 1992*; *Heitman et al., 1991*) and orthologous pathways were found in virtually all eukaryotes, including *Drosophila* (*Olsen et al., 1998*; *Zhang et al., 2002*). The TOR kinase forms two protein complexes: mTORC1 and mTORC2, which can be activated by many cues (*Laplante and Sabatini, 2009*). Especially, mTORC1 responds to high intracellular AA levels through the action of AA transporters and AA-binding cytosolic proteins; conversely, AA scarcity represses TOR activity (*Goberdhan et al., 2016*). Once activated, TOR increases translation through phosphorylation of 4E-BP and S6K (*Ma and Blenis, 2009*) and promotes systemic growth of *Drosophila* larvae (*Colombani et al., 2003*). GCN2 is activated by several cues (*Donnelly et al., 2013*, p. 2). The best-characterized cue is the quantity of uncharged tRNAs, a signature of hampered protein synthesis reflecting a scarcity in intracellular AA (*Masson, 2019*). Activation of GCN2 causes a global translational repression through phosphorylation of the eukaryotic initiation factor 2 (eIF2) (*Teske et al., 2011*) except for a subset of mRNAs which translation is increased. Among these mRNAs, transcription factors such as ATF4 then promote adaptation to stress (*Harding et al., 2003*). In addition to its cell-autonomous effects, the GCN2 pathway has systemic effects in *Drosophila*. Ubiquitous knock-down of GCN2 in *Drosophila* larvae causes developmental delay (*Malzer et al., 2013*). Moreover, GCN2 is involved in the regulation of other physiological processes. Activation of GCN2 in dopaminergic neurons of the larval brain (*Bjordal et al., 2014*) or in the enterocytes of adult flies (*Kim et al., 2021*) trigger a marked behavioural response leading to the avoidance of diets with an imbalanced AA composition. In addition, GCN2 in adult enterocytes regulates gut plasticity in response to AA/sugar imbalance (*Bonfini et al., 2021*). Furthermore, GCN2 in the midgut or in the fat body is necessary for lifespan extension under dietary restriction (*Kim et al., 2020*). Finally, GCN2 in the fat body represses reproduction under AA scarcity (*Armstrong et al., 2014*).

Although undernutrition greatly delays the development of germ-free (GF) *Drosophila* larvae (i.e. larvae lacking a microbiota), such delay can be buffered by the association of GF larvae with certain strains of symbiotic microbes (*Gould et al., 2018*; *Keebaugh et al., 2018*; *Shin et al., 2011*; *Storelli et al., 2011*). Growth promotion relies partly on nutrient provision: for instance, the symbiotic bacterium *Lactiplantibacillus plantarum* (formerly named *Lactobacillus plantarum* [*Zheng et al., 2020*], hereinafter referred to as Lp) can provide certain AA, which allows the larva to grow in the absence of these specific AA (*Consuegra et al., 2020b*). We sought to identify mechanisms of growth promotion that do not rely on AA provision. Therefore, we used a holidic diet (HD, diet composed of purified nutrients) (*Piper et al., 2017*) to create AA imbalance by specifically decreasing the quantity of an AA that Lp cannot synthesise. Interestingly, we found that Lp can still promote larval growth in these conditions. The mechanism thus does not rely on AA provision nor on stimulation of intestinal proteases (*Erkosar et al., 2015*) because the HD contains only free AA. Instead, we found that it depends on the production of ribosomal and transfer RNA (r/tRNA) by Lp, which are released in extracellular vesicles and lead to the activation of GCN2 in enterocytes. GCN2 activation results in a remodelling of the gut transcriptome towards epithelium maturation and altered metabolic activity, which may support systemic growth. Our study suggests that on top of its canonical role as a cellular sensor of uncharged eukaryotic tRNA, GCN2 is also a mediator of non-nutritional symbiotic cues encoded by bacterial r/tRNA operons.

## Results

### Association with Lp rescues AA imbalance

We used an HD which AA composition is based on *Drosophila*'s exome (FLY AA diet, *Piper et al., 2017*). This diet is well balanced in AA and allows optimal reproduction, lifespan, and growth. AA imbalance can be generated by decreasing the concentration of any essential AA (EAA), which then becomes limiting for animal fitness. Imbalanced diets cause defects in *Drosophila*'s growth, reproduction, and lifespan (*Piper et al., 2017*). In a previous study, we used HD to identify which AA can be synthesised by Lp and provided to the *Drosophila* larva. Lp cannot rescue the development of larvae on a diet that completely lacks an EAA that it cannot synthesise such as isoleucine, leucine, and valine (*Consuegra et al., 2020b*). Here, we wondered whether or not Lp could rescue the effects of AA imbalance due to limitations in EAA. To this end, we selectively decreased the concentration of each EAA by 70% from the FLY AA diet and measured the time of development of larvae (expressed as D50, i.e. the median time of entry into metamorphosis). As expected, decreasing the concentration of any EAA caused growth delay in GF larvae. Decreasing the amount of certain EAA (Ile, Lys, Thr, Trp, Val) completely blocked the development of GF larvae, which stalled at the L2 or L3 stage. Decreasing the amount of the other EAA (Arg, His, Met, Leu, Phe) caused an important developmental delay in GF larvae compared to the FLY AA diet (*Figure 1A*). In order to compare the different conditions, we aimed at finding a setup in which GF larvae are able to reach pupariation, with a strong developmental delay. We thus imposed a less severe AA imbalance by only decreasing the concentration of each EAA by 60%. We observed a similar trend: decreasing the amount of Ile, Lys, Thr, Trp, or Val yielded an important developmental delay in GF larvae, whereas decreasing the amount of Arg, His, Met, Leu, or Phe resulted in a minor developmental delay (*Figure 1B*). This confirms that some EAA seem to be more important than others for the development of GF larvae; alternatively, it is possible that the FLY AA diet does not contain enough of these EAA (Ile, Lys, Thr, Trp, Val) and too much of the others (Arg, His, Met, Leu, Phe).

Association with Lp rescued the effects of limitation in any EAA (*Figure 1A and B*). Lp's genome does not encode the enzymes necessary for the synthesis of branched-chained AA (BCAA: leucine, isoleucine, and valine) (*Martino et al., 2016*; *Saguir and de Nadra, 2007*; *Teusink et al., 2005*), and thus Lp cannot provide them to *Drosophila* (*Consuegra et al., 2020b*; *Kim et al., 2021*). Therefore, we were interested by the fact that Lp could rescue the developmental delay caused by BCAA limitation. We decided to focus on diets limited in valine (Val) to decipher the mechanisms underlying such beneficial effect of Lp on the growth of its host. We thereafter refer to the FLY AA diet as 'balanced diet', and to the FLY AA –60% Val diet as 'imbalanced diet'. Decreasing Val by 60% resulted in a strong delay in the growth of GF larvae, which was almost completely rescued by association with Lp (*Figure 1C*). Replacing the missing Val with an equal quantity of another EAA (Leu or His) did not improve the

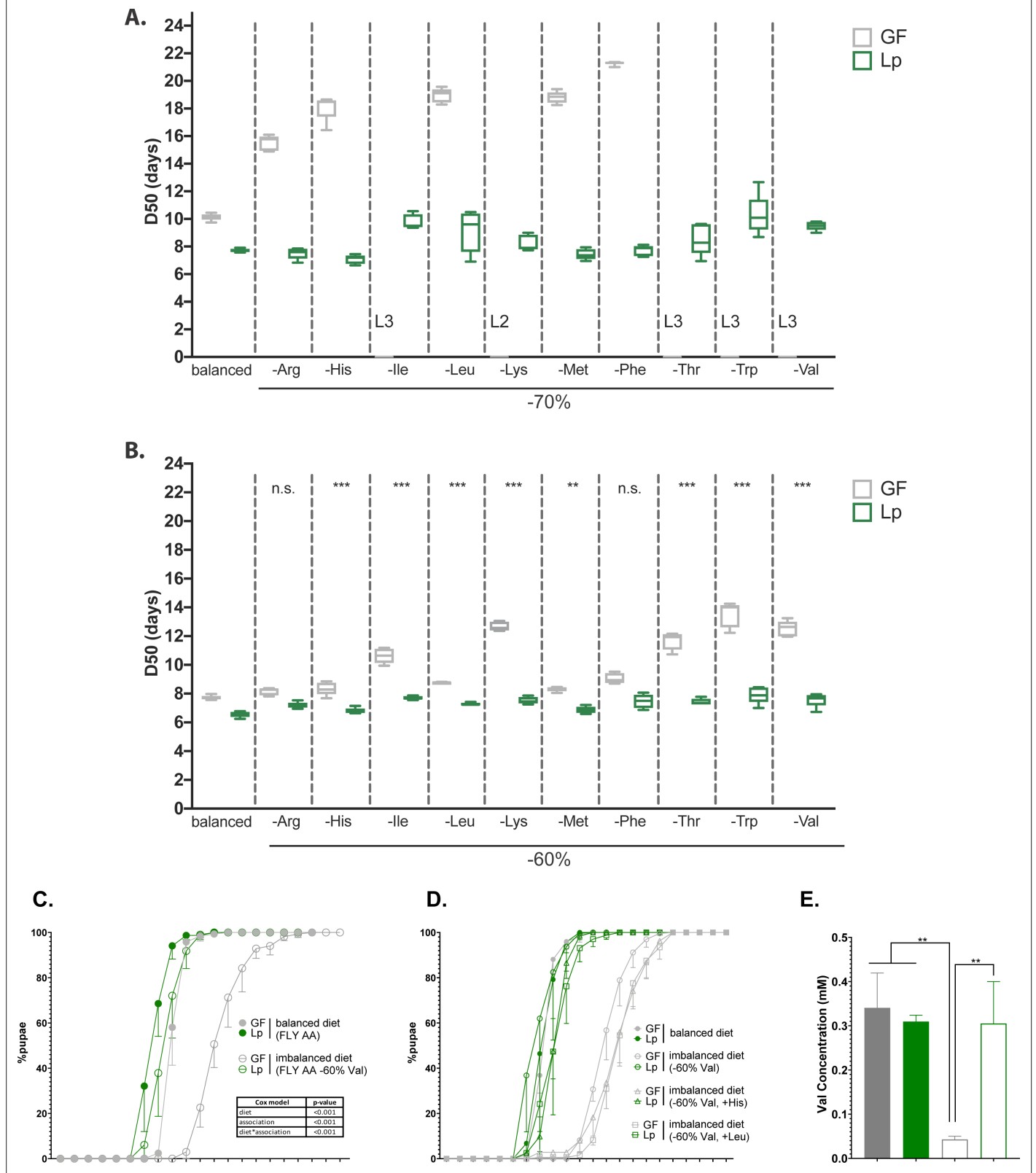

**Figure 1.** *L. plantarum* (Lp) rescues the developmental delay due to amino acid (AA) imbalance. (**A,B**) Developmental timing of germ-free (GF) larvae (grey) and Lp-associated larvae (green) on FLY AA diets with a –70% (**A**) or –60% (**B**) decrease in each essential AA (EAA). Boxplots show maximum, minimum, and median D50 (median time of pupariation) of five replicates. Each replicate consists in one tube containing 40 larvae. L2: larvae stalled at the L2 stage. L3: larvae stalled at the L3 stage. For each diet in (**B**), we used a Cox proportional hazards model to test the effect of the diet, association

*Figure 1 continued on next page*

*Figure 1 continued*

with Lp, and the interaction between these two parameters. We show the p-values of the interactions between diet and association with Lp after correction by the FDR method. n.s.: non-significant, **: p-value <0.01, ***: p-value <0.001. (**C**) Developmental timing of larvae raised on balanced diet (FLY AA, filled circles) or imbalanced diet (FLY AA –60% Val, empty circles). The graph represents the total fraction of emerged pupae over time as a percentage of the final number of pupae. The graph shows five replicates per condition (mean and standard deviation). Each replicate consists in one tube containing 40 larvae. We used a Cox proportional hazards model to test the effect of the diet, the association with Lp, and the interaction between these two parameters. (**D**) Developmental timing of larvae raised on balanced diet (FLY AA, filled circles), on imbalanced diet (FLY AA Val –60%, empty circles), on imbalanced diet adjusted with His (FLY AA Val –60%+His, triangles) or on imbalanced diet adjusted with Leu (FLY AA Val –60%+Leu, squares). The graph represents the total fraction of emerged pupae over time as a percentage of the final number of pupae. The graph shows five replicates per condition (mean and standard deviation). Each replicate consists in one tube containing 40 larvae. (**E**) Valine concentration in the haemolymph of larvae. The graph shows the mean and standard deviation of three replicates. Each replicate consists in the haemolymph of 10 larvae pooled together. We used an ANOVA followed by post hoc Dunnett's test to compare the mean of each condition to the mean of the condition GF on imbalanced diet. **: p-value <0.01.

The online version of this article includes the following source data and figure supplement(s) for figure 1:

**Source data 1.** Raw data displayed in *Figure 1*.

**Figure supplement 1.** Characterization of the growth promoting effect of *L. plantarum* on AA-imbalanced diet.

**Figure supplement 1—source data 1.** Raw data displayed in *Figure 1—figure supplement 1*.

---

development of GF larvae (*Figure 1D*). This shows that the delay observed on imbalanced diet is due to AA imbalance rather than total AA scarcity. Of note, further decreasing Val concentration (–80%, –90%) was lethal to GF larvae, but not to Lp-associated larvae (*Figure 1—figure supplement 1A*). Completely removing Val from the diet is lethal to both GF larvae and Lp-associated larvae as Lp is a Val auxotroph (*Consuegra et al., 2020b*). On the contrary, increasing Val by 100% compared to its initial levels did not impact the development of GF or Lp-associated larvae (*Figure 1—figure supplement 1B*). Moreover, egg-to-pupa survival was not impacted by AA imbalance nor by association with Lp (*Figure 1—figure supplement 1C*). Finally, supplementing the GF larvae with heat-killed (HK) Lp did not rescue the effects of an imbalanced diet on larval growth, which shows that the Val brought by the inoculation of Lp at the beginning of the experiment is not sufficient to restore Val levels required for larval growth (*Figure 1—figure supplement 1D*). Taken together, these results demonstrate that Lp can rescue the effects of AA imbalance due to a single dietary EAA limitation on larval growth through a mechanism independent of AA providing.

*Drosophila* can adapt their feeding behaviour in response to internal cues (*Bjordal et al., 2014*; *Gu et al., 2022*) and to their microbiota (*Leitão-Gonçalves et al., 2017*). We therefore tested whether GF larvae association with Lp altered larval food intake in our experimental conditions. We observed that Lp association does not alter food intake on imbalanced (*Figure 1—figure supplement 1E*) or balanced diets (*Figure 1—figure supplement 1F*). On a yeast-based diet, Lp stimulates the expression of intestinal proteases in larvae, which improves AA absorption by increasing the release of AA from dietary polypeptide (*Erkosar et al., 2015*). However, the HD that we used does not contain any polypeptide: its sole source of AA is free AA. We tested whether Lp could promote Val absorption in such conditions. We used high-pressure liquid chromatography (HPLC) to measure the Val concentration in the haemolymph of larvae. GF larvae fed an imbalanced diet (FLYAA –60% Val) showed a decrease in Val content in the hemolymph compared to GF larvae fed a balanced diet. This decrease was compensated upon association with Lp (*Figure 1E*). Therefore, we posit that Lp promotes physiological adaptation of its host to AA imbalance through increased AA absorption, which supports larval growth.

## Rescue of AA imbalance by Lp requires functional r/tRNAs operons in Lp

In order to decipher how Lp rescues the effects of AA imbalance on host growth, we performed a genetic screen using a transposon insertion library of Lp (*Figure 2A*). This library is composed of 2091 mutants, each carrying a transposon randomly inserted in the chromosome, including 1218 insertions inside open reading frames (*Matos et al., 2017*). We mono-associated GF larvae with each mutant of the library and looked for transposon insertions in Lp's genome altering the capacity of Lp to support larval development on a severely imbalanced diet (FLY AA –80% Val, *Figure 1—figure supplement 1A*). For each mutant, we calculated the D50 (median time of entry into metamorphosis)

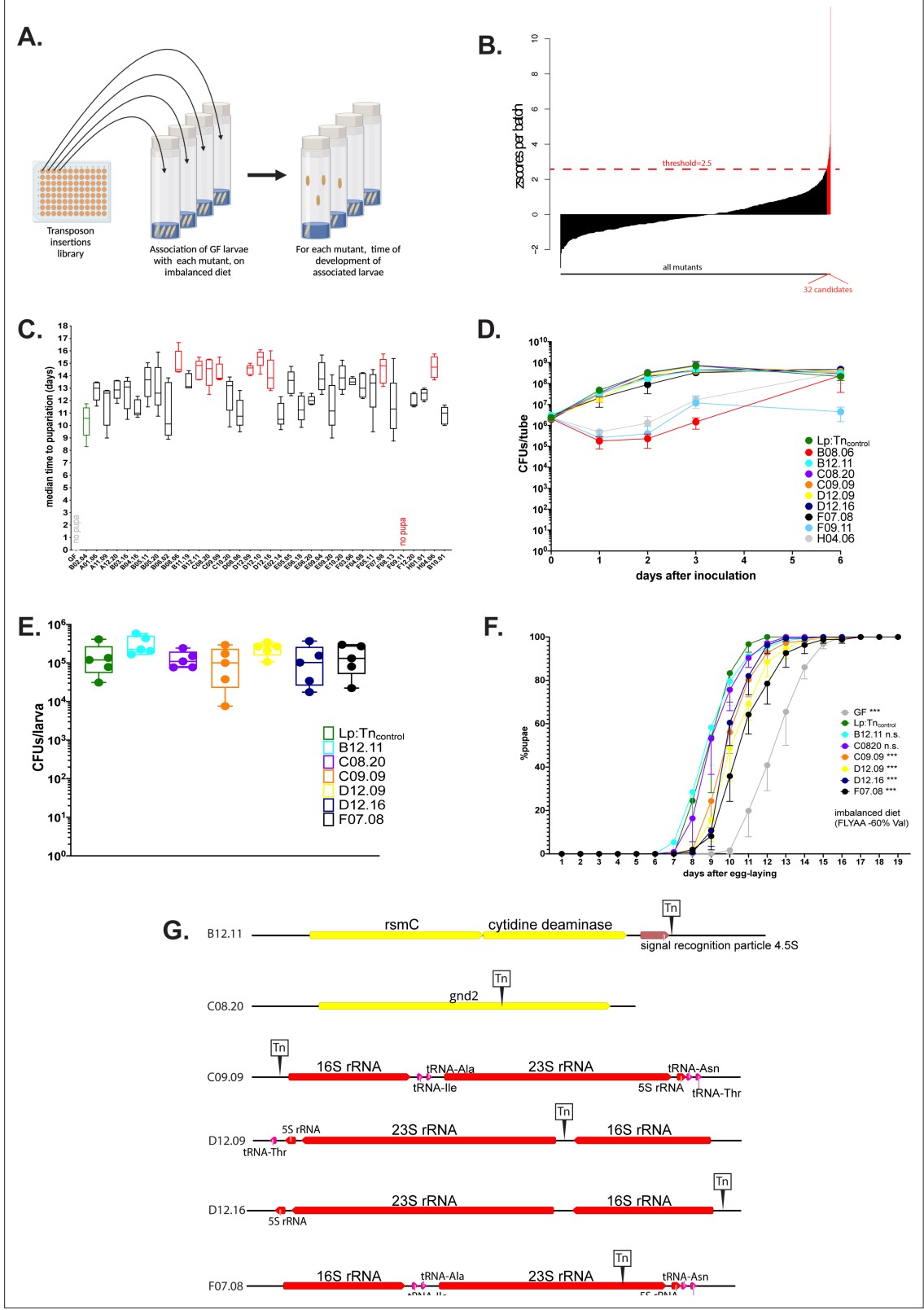

**Figure 2.** The operons encoding ribosomal and transfer RNAs (r/tRNAs) in *L. plantarum* (Lp) are necessary for Lp to rescue the delay due to amino acid (AA) imbalance. (**A**) Representation of the genetic screen. (**B**) Result of the screen: for each Lp mutant (X-axis), we calculated the median time of development of associated larvae on a severely imbalanced diet (FLY AA –80% Val) and normalised it into a z-score (Y-axis). We selected the 32 candidates that yielded a z-score >2.5. (**C**) Developmental timing of germ-free (GF) larvae (grey) and larvae associated with Lp:Tn$_{control}$ (green) or the

*Figure 2 continued on next page*

*Figure 2 continued*

32 candidate mutants from the genetic screen, on a severely imbalanced diet (FLY AA –80% Val). GF larvae and larvae associated with mutant F09.11 did not reach pupariation. Boxplots show maximum, minimum, and median of D50 (median time of pupariation) of five replicates per condition. Each replicate consists in one tube containing 40 larvae. We performed a Kruskal-Wallis test followed by post hoc Dunn's tests to compare all mutants to Lp:Tn_control. In red: statistically significant difference with Lp:Tn_control (p-value <0.05). (**D**) Growth of the nine candidates on imbalanced diet (FLY AA –60% Val), in association with larvae. The graph shows the quantity of colony-forming units (CFUs) of Lp over time (mean and standard deviation of three replicates). (**E**) Colonisation of the larval gut by the six remaining candidates, on imbalanced diet (FLY AA –60% Val). The graph shows the quantity of CFUs of Lp per larva (mean and standard deviation of 5 replicates). We performed a Kruskal-Wallis test followed by post hoc Dunn's tests to compare each candidate to Lp:Tn_control and found no statistically significant difference. (**F**) Developmental timing of larvae raised on imbalanced diet (FLY AA –60% Val), in GF condition or in association with each one of the six candidates or with Lp:Tn_control. The graph represents the total fraction of emerged pupae over time as a percentage of the final number of pupae. The graph shows five replicates per condition (mean and standard deviation). Each replicate consists in one tube containing 40 larvae. We used a Cox proportional hazards model to compare the effect of each candidate to the effect of Lp:Tn_control. The p-values were adjusted by the Tukey method. n.s.: non-significant. ***: p-value <0.001. (**G**) Representation of the six transposon insertions. Tn: transposon. rspC: 16S rRNA methyltransferase. gnd2: phosphogluconate dehydrogenase. Of note, C09.09 and F07.08 show two independent insertions in the same r/tRNA operon.

The online version of this article includes the following source data and figure supplement(s) for figure 2:

**Source data 1.** Raw data displayed in *Figure 2*.

**Figure supplement 1.** *L. plantarum* mutants for r/tRNA operon show a decrease in the abundance of r/tRNAs, but no effect on growth or host colonization.

**Figure supplement 1—source data 1.** Raw data displayed in *Figure 2—figure supplement 1*.

and normalised it into a z-score. We applied a threshold of z-score >2.5 and identified 32 insertional mutants. Association with these mutants thus results in a delayed time of larval development on a severely imbalanced diet (*Figure 2B*). To validate these 32 candidates, we individually re-tested them in multiple (5) replicates. We compared the development of larvae associated with the 32 candidates to larvae associated with an intergenic region insertion mutant of the library showing a WT-like phenotype (Lp:Tn_control, z-score=0.65). Thus, we discarded 23 false positives and retained only 9 candidates which resulted in a significant and robust developmental delay on an imbalanced diet upon association (*Figure 2C*).

Upon association with *Drosophila* larvae Lp grows on the fly food and constantly and repetitively transits through the larval gut (*Storelli et al., 2018*). The quantity of live bacteria present in the food can greatly impact the growth-promoting capacity of Lp (*Consuegra et al., 2020a*; *Keebaugh et al., 2018*). As a consequence, an Lp strain that grows poorly on the food matrix would not support larval growth. Since we wanted to exclude such candidates, we tested the growth of the nine candidates on imbalanced HD, in the presence of larvae (*Figure 2D*). Three candidates (B08.06, F09.11, and H04.06) showed growth defects on the diet and thus were not retained for further analysis. On the contrary, the remaining six candidates (B12.11, C08.20, C09.09, D12.09, D12.16, F07.08) showed no growth defect. Moreover, they did not show any impairment at colonising the larval gut (*Figure 2E*). For further characterisation, we tested whether the effect of the mutations could also be observed on a moderately imbalanced diet (–60% Val). On such diet, the mutations B12.11 and C08.20 did not significantly impact Lp's ability to rescue the developmental delays of larvae. On the contrary, larvae associated with the mutants C09.09, D12.09, D12.16, and F07.08 were still delayed compared to the larvae associated with the WT-like mutant Lp:Tn_control (*Figure 2F*), though the difference was less important than on a severely imbalanced diet (–80% Val, *Figure 2C*).

Next, we sequenced the genomes of the six selected candidates to determine in which genomic regions the transposons were inserted. Interestingly, four out of the six candidates (mutants C09.09, D12.09, D12.16, and F07.08) showed independent transposon insertions in operons containing genes encoding tRNAs and/or rRNAs (*Figure 2G*). These four mutants are also the ones that seem to most affect larval development on imbalanced diet (*Figure 2F*). The genome of Lp contains five operons encoding r/tRNAs. C09.09 and F07.08 display independent insertions in the same operon, but at different loci (upstream the 16S rRNA for C09.09 and inside the 23S rRNA for F07.08, *Figure 2G*). We further characterised F07.08, hereinafter referred to as Lp:Tn_r/tRNA. We extracted total RNA from Lp cultured in liquid HD and found that despite the redundancy of r/tRNAs operon in Lp's genome Lp:Tn_r/tRNA displays an ~3- to 4-fold reduction of 16S (*Figure 2—figure supplement 1A*) and 23S

(*Figure 2—figure supplement 1B*) rRNA compared to Lp:Tn_control, indicating that altering the sequence of a single r/tRNA operon is sufficient to compromise global rRNA levels in Lp cells without altering cell growth.

To confirm the importance of r/tRNA operons in supporting larval growth, we generated a deletion mutant of the whole operon identified in the insertion mutants C09.09 and Lp:Tn_{r/tRNA} by homology-based recombination (*Matos et al., 2017*). The deletion mutant (hereinafter referred to as LpΔop_{r/tRNA}) showed the same phenotype as the insertion mutants in terms of lack of rescue of AA imbalance (–60% Val, *Figure 2—figure supplement 1C*; –70% Leu, *Figure 2—figure supplement 1D*), growth on imbalanced HD in the presence of larvae (*Figure 2—figure supplement 1E*), larval gut colonisation (*Figure 2—figure supplement 1F*), and reduction of the expression of the 16S (*Figure 2—figure supplement 1G*) and 23S (*Figure 2—figure supplement 1H*) rRNA (~3-fold reduction).

In *Escherichia coli*, operons encoding r/tRNA operons can also encode small RNAs (sRNAs) (*Stenum et al., 2021*). Bacterial sRNAs are regulatory RNAs belonging to the general class of non-coding RNAs (ncRNAs). They can be a major mediator or host-symbiont interactions: for instance, the squid symbiont *Vibrio fischeri* produces the sRNA SsrA, which promotes immune tolerance from its host (*Moriano-Gutierrez et al., 2020*). Therefore, we purified and sequenced the ncRNAs expressed by LpWT and LpΔop_{r/tRNA} in culture. We identified 13 ncRNAs other than r/tRNAs (*Supplementary file 1*). Of those, 10 were encoded by leading sequences within the 5'-UTR of mRNAs, and are annotated as translation regulators. The remaining 3 ncRNAs (RNAseP, SRP4.5S, and tmRNA, which is the homolog of SsrA) are bona fide sRNAs but they are not encoded in the r/tRNA operons. In addition, we used our sRNAseq dataset to quantify the levels of each tRNAs. As a proxy for tRNA quantity, we focused on Thr-tRNAs. There are four copies of Thr-tRNAs in Lp's genome: tRNA05, which is inside the operon that was deleted in LpΔop_{r/tRNA} (*Figure 2G*), and tRNA15, tRNA57, and tRNA69, which are outside the operon. The sequences of these four copies are very similar but we were able to assign certain uniquely mapped reads from our dataset to each specific copy using single nucleotide polymorphisms and the flanking regions. As expected, the number of reads mapping specifically on Thr-tRNA5 (inside the operon) was decreased >99% in LpΔop_{r/tRNA} compared to Lp. On the other hand, the three Thr-tRNAs outside the deleted operon did not show any difference of expression (*Figure 2—figure supplement 1I*). This shows that the deletion of one copy of Thr-tRNA did not induce a compensatory increase of the other copies. The fact that the chromosome of Lp contains five operons encoding rRNAs and multiple copies of each tRNA may explain why deleting one r/tRNA operon does not have a major impact on bacterial fitness (*Figure 2D and E*; *Figure 2—figure supplement 1E and F*). However, it appears that the production of r/tRNAs from these operons is rate limiting to support *Drosophila* growth upon AA imbalance.

## Lp produces extracellular vesicles containing r/tRNAs

RNA produced by bacteria can be sensed by eukaryotic host's cells and modulate host's immune signalling (*Oldenburg et al., 2012*, p. 13; *Ren et al., 2019*). We hypothesised that products of the r/tRNAs loci may be microbial cues sensed by host cells to promote physiological adaptation. Lp is present in the endoperitrophic space of the gut: it is not in direct contact with the enterocytes (*Storelli et al., 2018*). Therefore, we wondered how r/tRNAs may be transferred from bacteria to host cells. sRNAs from *Pseudomonas aeruginosa* (*Koeppen et al., 2016*) and *V. fischeri* (*Moriano-Gutierrez et al., 2020*) were found inside extracellular vesicles. *Lacticaseibacillus casei* produces extracellular vesicles that contain r/tRNAs (*Domínguez Rubio et al., 2017*). Moreover, extracellular vesicles from *Limosilactobacillus reuteri* influence gut motility in mice (*West et al., 2020*) and extracellular vesicles from Lp^WCFS1 increase the expression of immunity genes in worms and cultured colonocytes (*Li et al., 2017*). We thus hypothesised that extracellular vesicles may act as 'vehicles' allowing transfer of Lp's r/tRNAs from bacteria to host cells. We isolated extracellular vesicles from Lp's supernatant and imaged them using electron microscopy, revealing the presence of spherical structures of 70–200 nm of size (*Figure 3A*). This corresponds to the expected size of Lp's extracellular vesicles (*Li et al., 2017*). We then extracted RNA from the vesicles and from bacterial cells and quantified the amount of rRNA and tRNA by RT-qPCR. In cells, 16S rRNA and 23S rRNA were considerably more abundant than Thr-tRNA, as expected from previous studies (*Giannoukos et al., 2012*; *Figure 3B*). In purified vesicles, we observed an enrichment of Thr-tRNAs (*Figure 3C*). In order to remove potential contaminants from the supernatant and retain only the RNAs present inside the vesicles, we treated the purified vesicles

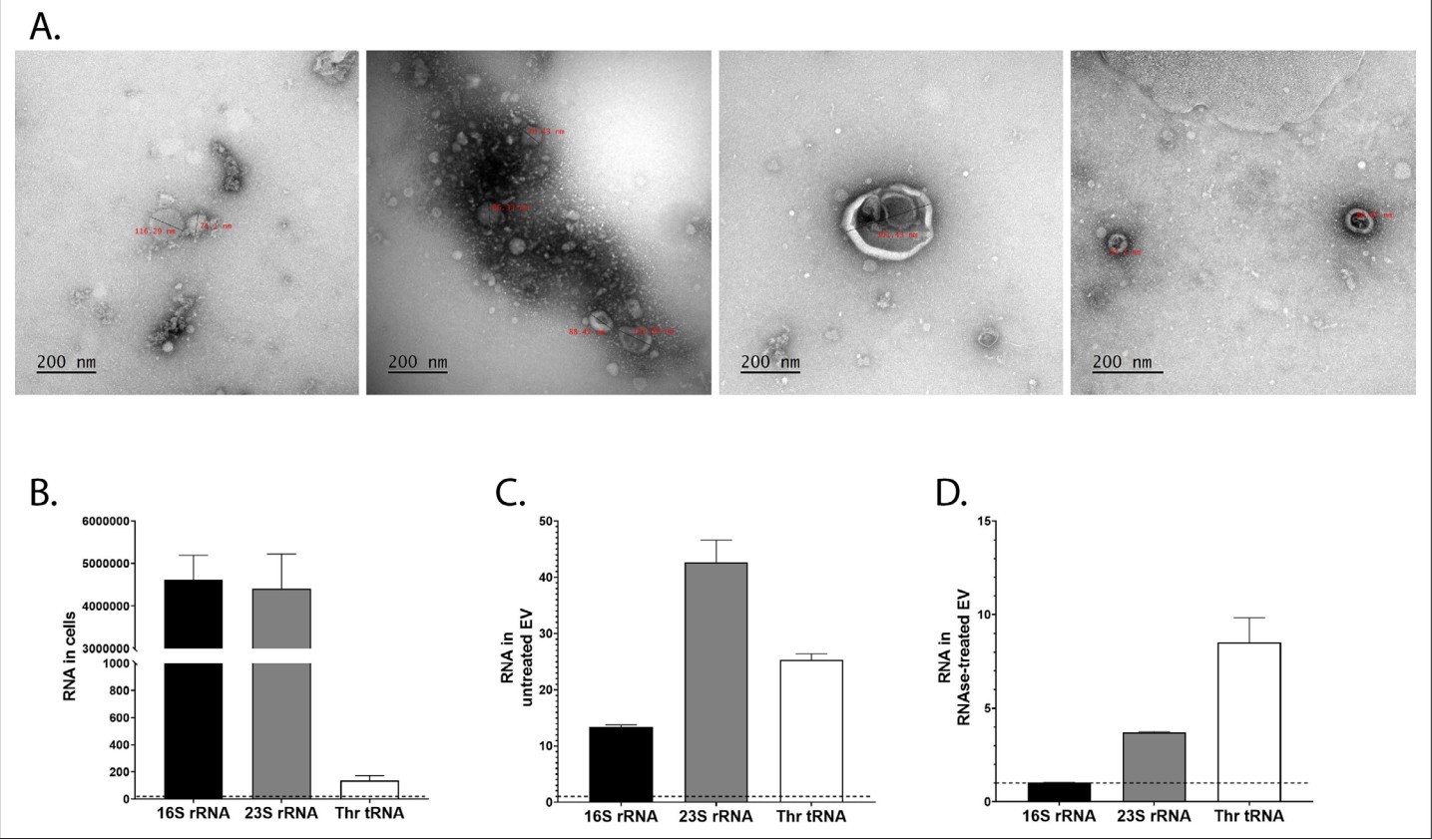

**Figure 3.** *L. plantarum* (Lp) produces extracellular vesicles containing ribosomal and transfer RNAs (r/tRNAs). (**A**) Representative transmission electronic microscopy images of extracellular vesicles purified from Lp's supernatant. The text in red shows the diameter in nm. The scale bar indicates 200 nm. (**B**–**D**) RT-qPCR quantification of 16S rRNA, 23S rRNA, and Thr-tRNA from total bacterial cells (**B**), purified extracellular vesicles (**C**), or purified extracellular vesicles treated with RNAse (**C**), normalised with sterile supernatant. RNA quantity is shown as $2^{Cq(\text{sterile supernatant})-Cq(\text{cells})}$ or $2^{Cq(\text{sterile supernatant})-Cq(\text{extracellular vesicles})}$. The dotted line shows the level of RNA detected in sterile supernatant. The graph shows the mean of two replicates and standard deviation.

The online version of this article includes the following source data for figure 3:

**Source data 1.** Raw data displayed in *Figure 3*.

with RNAse before extracting RNA. Most rRNA were depleted by the RNAse treatment, suggesting that they were contaminants from the supernatant. On the contrary, Thr-tRNAs were enriched after RNAse treatment (*Figure 3C*). These results suggest that Lp's extracellular vesicles contain mostly tRNAs, which may allow their transfer to host cells.

## Lp activates GCN2 signalling in the anterior midgut

We next wondered whether Lp's r/tRNA can be sensed by host cells. The GCN2 kinase can bind to and be activated by unloaded tRNAs (*Masson, 2019*, p. 2) and rRNAs (*Zhu and Wek, 1998*). Moreover, the GCN2 pathway is one of the main pathways that allow eukaryotic cells to adapt to AA imbalance (*Gallinetti et al., 2013*, p. 2). Finally, GCN2 is active in the gut of *Drosophila* (*Bonfini et al., 2021*; *Kim et al., 2021*), where we observed Lp in close proximity to host's cells (*Figure 4—figure supplement 1A*). Therefore, we wondered if GCN2 may be a sensor and signalling intermediary between r/tRNA from Lp and host physiological adaptation to an AA-imbalanced diet.

We first tested whether Lp association regulates GCN2 activation in the larval gut. In *Drosophila*, the transcription factor ATF4 acts downstream of GCN2 and binds to recognition sites in the first intron of the gene *4E-BP* (*Thor* in *Drosophila*) to activate its transcription (*Kang et al., 2017*). Kang and colleagues previously generated a transgenic line (4E-BP^intron^dsRed), which carries a fluorophore under the transcriptional control of the first intron of *4E-BP*. This reporter thus allows to visualise the pattern of activity of ATF4 downstream of GCN2 activation (*Kang et al., 2017*; *Vasudevan et al., 2017*). We used the 4E-BP^intron^dsRed reporter as a molecular readout to probe GCN2 activity in the

larval midgut in response to Lp association. *Figure 4—figure supplement 1B* shows the pattern of expression of the 4E-BP$^{intron}$dsRed reporter in dissected guts of larvae fed an imbalanced diet (top panel) or a balanced diet (bottom panel), either GF (left panel) or Lp-associated (right panel). Similarly to what was previously reported (*Kang et al., 2017*), we observed 4E-BP$^{intron}$dsRed reporter expression in the gastric caeca, the proventriculus and in the middle midgut, in a region known as the acidic zone (*Overend et al., 2016*). This pattern was conserved between GF larvae and Lp-associated larvae. Conversely, the 4E-BP$^{intron}$dsRed reporter was expressed in the anterior midgut specifically in Lp-associated larvae, while this signal was absent from GF guts (*Figure 4—figure supplement 1B*, red squares, and *Figure 4A*). We confirmed by RT-qPCR that endogenous GCN2-dependent *4E-BP* expression is induced in the anterior midgut upon Lp-association on an imbalanced diet (*Figure 4—figure supplement 1C*). Interestingly, expression of the 4E-BP$^{intron}$dsRed reporter in this region depends on the association with Lp, but not on AA imbalance as we observed it in larvae raised on either imbalanced diet (*Figure 4A*, top panel) or balanced diet (*Figure 4A*, bottom panel; quantification of the signal is shown in *Figure 4B*). Lp can thus activate the 4E-BP$^{intron}$dsRed reporter specifically in the anterior midgut, independently of dietary AA imbalance.

ATF4 is activated by eIF2, which can be phosphorylated by GCN2 but also by other kinases such as PERK (*Teske et al., 2011*). In order to test whether the 4E-BP$^{intron}$dsRed reporter indeed mirrors GCN2 activity, we looked at its pattern of expression in a GCN2 knock-down background. Inhibition of GCN2 expression using tissue-specific in vivo RNAi (*Dietzl et al., 2007*) completely abrogated the activation of the 4E-BP$^{intron}$dsRed reporter by Lp in the anterior midgut of larvae reared on imbalanced diet (*Figure 4C and D*) or balanced diet (*Figure 4E and F*). Therefore, our results establish that Lp promotes GCN2 activity in the anterior midgut of larvae, independently of AA imbalance.

## Lp mutants for r/tRNA operon fail to activate GCN2 in the anterior midgut

We then tested whether activation of GCN2 by Lp depends on its r/tRNA loci. To this end, we compared the induction of the 4E-BP$^{intron}$dsRed reporter in larvae associated with LpΔop$_{r/tRNA}$ and in larvae associated with Lp WT. On an imbalanced diet, LpΔop$_{r/tRNA}$-associated larvae showed a reduced activation of GCN2 compared to WT-associated larvae, although this reduction did not reach statistical significance (*Figure 5A and B*). On a balanced diet, GCN2 activation in LpΔop$_{r/tRNA}$-associated larvae was comparable to GCN2 activation in GF larvae, and significantly reduced compared to GCN2 activation in Lp WT-associated larvae (*Figure 5C and D*). Similarly, we observed that association with Lp:Tn$_{r/tRNA}$ yielded a reduction of GCN2 activation comparable to what we observed with LpΔop$_{r/tRNA}$ (on imbalanced diet: *Figure 5—figure supplement 1A and B*; on balanced diet: *Figure 5—figure supplement 1C and D*). The lesser difference observed on imbalanced diet between Lp:Tn$_{control}$-associated larvae and Lp:Tn$_{r/tRNA}$-associated larvae might be due to longer association of the larvae with Lp:Tn$_{r/tRNA}$: indeed, in order to size-match the larvae, we collected them 24 hr before the emergence of the first pupae, which is D6 after egg-laying (AEL) for Lp:Tn$_{control}$ and D8 AEL for Lp:Tn$_{r/tRNA}$. To ensure comparable association time with the two mutants, we thus performed short-term association of GF larvae with Lp:Tn$_{control}$ or Lp:Tn$_{r/tRNA}$: the larvae were reared GF, associated with Lp:Tn$_{control}$ or Lp:Tn$_{r/tRNA}$ at D8 AEL, and collected for dissection at D10 AEL. In short-term association on an imbalanced diet, activation of GCN2 was significantly reduced in Lp:Tn$_{r/tRNA}$-associated larvae compared to Lp:Tn$_{control}$-associated larvae (*Figure 5—figure supplement 1E and F*).

Uncharged tRNAs and rRNAs from eukaryotic cells are canonical activators of GCN2 kinase (*Masson, 2019*, p. 2; *Zhu and Wek, 1998*). It is currently unknown whether bacterial r/tRNAs can activate GCN2 as well. We therefore tested this hypothesis by feeding purified bacterial tRNAs to GF larvae carrying the 4E-BP$^{intron}$dsRed reporter. At the highest dose tested (625 μg), bacterial tRNAs significantly increased the expression of the reporter in the anterior midgut (*Figure 5E and F*). This increase was comparable to the effect of feeding eukaryotic tRNAs to these larvae, though slightly inferior. However, the effect was minimal as compared to the association of larvae with Lp (*Figure 5—figure supplement 1G*). Of note, Lp reaches a maximum of ~2.5 × 10⁹ CFUs·mL⁻¹ in HD (*Figure 2D*). Based on observations made on other bacteria, this may yield ~225 μg of tRNAs in a 10 mL vial (*Battley, 1988*). This value is probably an underestimate, because tRNAs may accumulate during the stationary phase of Lp; the value of 625 μg thus appears to be within the physiological range of exposure of larvae. These results therefore suggest that Lp's tRNA may be direct activators of GCN2

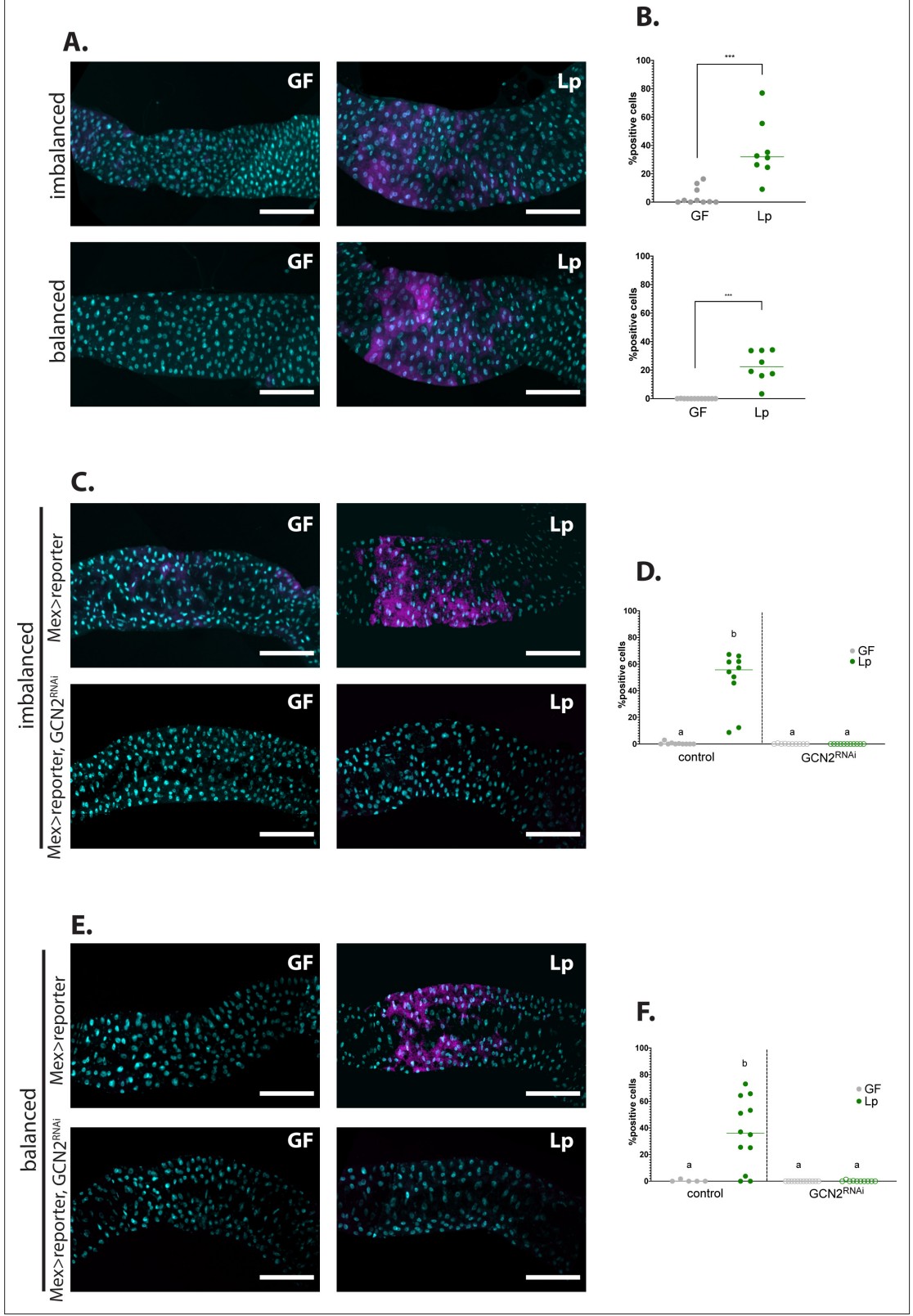

**Figure 4.** Association with *L. plantarum* (Lp) activates general control nonderepressible 2 (GCN2) in the larval anterior midgut. (**A,C,E**) Representative pictures of the anterior midgut of 4E-BP$^{intron}$dsRed larvae. Cyan: DAPI. Magenta: 4E-BP$^{intron}$dsRed reporter. Scale bar: 200 μm. (**B,D,F**) Quantification of the signal: proportion of enterocytes positive for 4E-BP$^{intron}$dsRed reporter's activity in the anterior midgut. Each dot represents an independent replicate. The bar shows the mean. (**A**) Larvae in germ-free (GF) conditions (left) or Lp-associated conditions (right) on imbalanced (FLY AA –60% Val,

*Figure 4 continued on next page*

*Figure 4 continued*

top) or balanced (FLY AA, bottom) diet. Quantification of the signal in (**B**): GF larvae (grey) or Lp-associated larvae (green) fed an imbalanced (top) or balanced (bottom) diet. We performed a Mann-Whitney test to compare the GF and Lp-associated conditions. **: p-value <0.01. ***: p-value <0.001. (**C–E**) Control larvae (Mex-Gal4x4E-BP$^{intron}$dsRed reporter) (top panel) and GCN2 knock-down (Mex-Gal4x4E-BP$^{intron}$dsRed reporter, UAS-GCN2$^{RNAi}$) (bottom panel) in GF conditions (left) and Lp-associated conditions (right) fed an imbalanced diet (FLY AA –60% Val) (**C**) or a balanced diet (FLY AA) (**E**). Quantification of the signal in (**D,F**): GF larvae (grey circles) or Lp-associated larvae (green circles) fed an imbalanced diet (**D**) or a balanced diet (**F**). Filled circles: control condition (Mex-Gal4x4E-BP$^{intron}$dsRed reporter). Empty circles: GCN2 knock-down (Mex-Gal4x4E-BP$^{intron}$dsRed reporter, UAS-GCN2$^{RNAi}$). Each dot represents one larva. We performed a Kruskal-Wallis test followed by post hoc Dunn's tests to compare all conditions together. a: the conditions are not significantly different from each other (p-value >0.05). b: the condition is significantly different from other conditions (p-value <0.01).

The online version of this article includes the following source data and figure supplement(s) for figure 4:

**Source data 1.** Raw data displayed in *Figure 4*.

**Figure supplement 1.** *L. plantarum* activates GCN2 in the larval anterior midgut.

**Figure supplement 1—source data 1.** Raw data displayed in *Figure 4—figure supplement 1*.

---

in enterocytes. However, we cannot exclude that Lp's rRNAs or an indirect mechanism dependent on a functional Lp r/tRNA operon is at play.

## Rescue of AA imbalance by Lp requires GCN2 in the larval midgut

We showed that association with Lp activates GCN2 in the anterior midgut of larvae in an r/tRNA loci-dependent manner. We thus sought to test whether GCN2 activation in enterocytes is necessary for Lp to rescue the developmental delay due to AA imbalance. To this end, we specifically knocked down GCN2 in enterocytes and followed the developmental timing of GF or Lp-associated larvae in AA-imbalanced conditions. On an imbalanced diet (–60% Val), GCN2 knock-down in enterocytes caused a significant developmental delay in Lp-associated larvae compared to control larvae (*Figure 6A*), a phenotype that we confirmed by using two other GCN2 RNAi lines (*Figure 6—figure supplement 1A and B*) and by measuring the efficacy of RNA interference against GCN2 expression in the midgut (*Figure 6—figure supplement 1C, D*). Of note, GCN2 knock-down in enterocytes does not alter colonisation of the gut by Lp (*Figure 6—figure supplement 1E*). Knocking-down GCN2 in the gut had no effect on larvae developing on an AA-balanced diet (*Figure 6B*). Taken together, our results indicate that GCN2 is necessary for Lp to rescue the developmental delay due to Val limitation. We wondered whether GCN2 is necessary to rescue AA imbalance due to limitation in other AA. We decreased the amount of each EAA identified in *Figure 1* as most important for GF larvae (Ile, Lys, Thr, and Trp) by 60% and measured the growth of larvae knocked down for GCN2. We found that GCN2 is necessary for Lp to rescue development upon scarcity in Ile or Thr, but not in Trp or Lys (*Figure 6C–F*). Lp can synthesise Lys and Trp, but not Ile (*Consuegra et al., 2020b*); therefore, it seems that GCN2 is necessary only when Lp cannot provide the limiting AA. Lp can produce Thr, but in limiting quantity (*Consuegra et al., 2020b*), which may explain why GCN2 is also necessary for Lp to rescue the delay due to Thr scarcity.

Lab-reared *Drosophila* are mostly associated with bacteria belonging to the families Lactobacillales and Acetobacteraceae (*Staubach et al., 2013*). We tested whether a representative Acetobacteraceae strain, *A. pomorum*$^{WJL}$ (Ap) that promotes larval growth on oligidic nutrient-poor diet (*Shin et al., 2011*) and on complete HD (*Consuegra et al., 2020a*), displays the same properties as Lp on AA-imbalanced HD. Ap rescued the developmental delay due to AA imbalance to the same extent as Lp (*Figure 6—figure supplement 2A*). However, conversely to Lp, Ap's support to larval development upon Val limitation was independent of GCN2 expression in the gut (*Figure 6—figure supplement 2B*). This is likely explained by the ability of Ap to produce Val and rescue the host's auxotrophy to Val (*Consuegra et al., 2020b*). Interestingly, Ap did not activate GCN2 in the anterior midgut of larvae on imbalanced diet (*Figure 6—figure supplement 2C and D*) or balanced diet (*Figure 6—figure supplement 2E and F*). Therefore, the ability to activate GCN2 in the anterior midgut is not common to all symbiotic bacteria and a specific attribute of Lp-mediated benefit.

In *Drosophila*, it was previously reported that AA scarcity engages the GCN2 pathway in enterocytes of the midgut (*Bonfini et al., 2021*; *Kim et al., 2021*) but also in fat body cells (*Armstrong et al., 2014*). We used the driver C564-Gal4 to knock down GCN2 expression in the fat body as well as other tissues such as salivary glands and hemocytes (*Takehana et al., 2004*). This had no impact

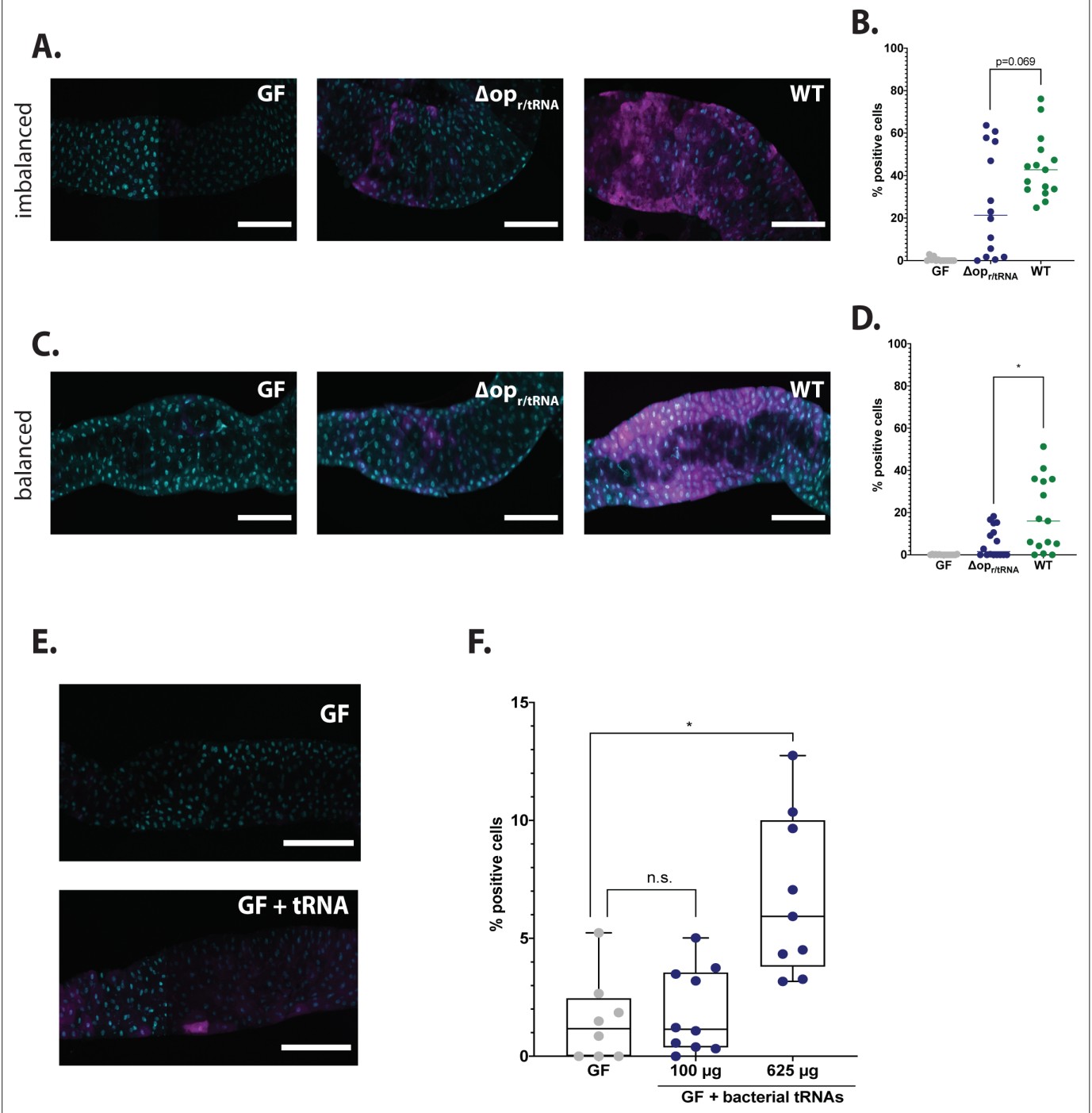

**Figure 5.** *L. plantarum* (Lp) mutants for ribosomal and transfer RNA (r/tRNA) operon fail to activate general control nonderepressible 2 (GCN2) in the anterior midgut. (**A,C**) Representative pictures of the anterior midgut of 4E-BP$^{intron}$dsRed larvae in germ-free (GF) conditions (left panels), in association with LpΔop$_{r/tRNA}$ (middle panels) or in association with Lp WT (right panels). Cyan: DAPI. Magenta: 4E-BP$^{intron}$dsRed reporter. Scale bar: 200 μm. (**B,D**) Quantification of the signal: proportion of enterocytes positive for 4E-BP$^{intron}$dsRed reporter's activity in the anterior midgut of GF larvae (grey), larvae associated with LpΔop$_{r/tRNA}$ (blue) and larvae associated with Lp WT (green). Each dot represents one larva. The bar shows the mean. We performed Mann-Whitney test to compare association with Lp WT and association with LpΔop$_{r/tRNA}$. *: p-value <0.05. (**A**) Larvae fed an imbalanced diet (FLY AA –60% Val), signal quantified in (**B**). (**C**) Larvae fed a balanced diet (FLY AA), signal quantified in (**D**). (**E**) Representative images of 4E-BP$^{intron}$dsRed GF larvae (top panel) and 4E-BP$^{intron}$dsRed GF larvae fed 625 μg of bacterial tRNAs (bottom panel) on balanced diet (FLY AA). Cyan: DAPI. Magenta: 4E-BP$^{intron}$dsRed reporter. (**F**) Quantification of the signal: proportion of enterocytes positive for 4E-BP$^{intron}$dsRed reporter's activity in the anterior midgut of GF larvae (grey), GF larvae fed with increasing concentrations of bacterial tRNAs (blue) on balanced diet (FLY AA). Each dot represents one larva. We performed a Kruskal-Wallis test followed by post hoc Dunn's tests to compare each condition to GF. n.s.: non-significant. *: p-value <0.05.

*Figure 5 continued on next page*

*Figure 5 continued*

The online version of this article includes the following source data and figure supplement(s) for figure 5:

**Source data 1.** Raw data displayed in *Figure 5*.

**Figure supplement 1.** The *L. plantarum* (Lp) insertion mutants for ribosomal and transfer RNA (r/tRNA) operon identified in the genetic screen fail to activate general control nonderepressible 2 (GCN2) in the anterior midgut.

**Figure supplement 1—source data 1.** Raw data displayed in *Figure 5—figure supplement 1*.

on the development of Lp-associated larvae fed an imbalanced diet (*Figure 6—figure supplement 1F*), which suggests that the phenotype that we observed engages GCN2 in enterocytes rather than in other tissues. Of note, although the TOR pathway is another major factor supporting systemic host growth in the fat body (*Colombani et al., 2003*) and in enterocytes (*Redhai et al., 2020*), it was not required in the fat body (*Figure 6—figure supplement 1G*) nor in enterocytes (*Figure 6—figure supplement 1H*) for Lp to rescue AA imbalance. However, it was important to support the growth of GF animals (*Figure 6—figure supplement 1G and H*).

Upon activation, GCN2 phosphorylates eIF2, the major subunit of the translation initiation factor in eukaryotic cells (*Donnelly et al., 2013*, p. 2). Phosphorylation of eIF2 inhibits translation, except for a subset of mRNAs including the transcription factor ATF4 which translation is increased (*B'chir et al., 2013*). ATF4 then activates the transcription of genes involved in stress response. In *Drosophila*, one of these genes is the eIF4E-binding protein 4E-BP (*Kang et al., 2017*). 4E-BP activation promotes cap-independent translation, which boosts anti-microbial peptides (AMPs) production in the fat body (*Vasudevan et al., 2017*, p. 2039-2047) and can alter the composition of the microbiota (*Vandehoef et al., 2020*). Therefore, we tested whether ATF4 and 4E-BP acting downstream of GCN2 are necessary for Lp to support larval development upon dietary AA imbalance. We observed that knocking down ATF4 or 4E-BP in the enterocytes delayed the development of GF larvae on an imbalanced diet. However, such knock-downs did not affect the development of Lp-associated larvae like GCN2 knock-down did. ATF4 and 4E-BP are thus not required for Lp to support larval development upon dietary AA imbalance (*Figure 6G*). Similarly, larvae carrying 4E-BP deletions were not delayed in presence of Lp. On the contrary, they grew slightly faster than Lp-associated yw larvae (*Figure 6—figure supplement 1I*).

Taken together, our results demonstrate a specific role of the GCN2 pathway in enterocytes to mediate Lp's support to larval development despite dietary AA imbalance, independently of TOR and ATF4.

We then tested the interaction between GCN2 knock-down and Lp r/tRNA loci insertion mutants. As expected, control larvae reared on imbalanced diet in association with Lp:Tn$_{r/tRNA}$ showed a developmental delay compared to larvae associated with Lp:Tn$_{control}$ (*Figure 6H*). On the contrary, larvae knocked-down for GCN2 in the enterocytes did not show a difference between association with Lp:Tn$_{r/tRNA}$ and association with Lp:Tn$_{control}$ (*Figure 6I*); in other words, the effect of the r/tRNA operon mutation was only observed when GCN2 is fully expressed. This suggests a functional interaction between Lp's r/tRNAs and GCN2 in the larval gut and supports the notion that Lp's influence on host adaptation to AA imbalance relies on a molecular dialogue engaging symbiont r/tRNAs and the host GCN2 kinase.

## The anterior midgut of Lp-associated larvae displays GCN2-dependent, ATF4-independent, and r/tRNA-dependent transcriptomic signatures of increased epithelium maturation and altered metabolic activity

We next wondered how GCN2 activation by products of Lp's r/tRNA operon may support larval growth on an AA imbalanced diet. To get a first handle on the molecular roots of this phenomenon, we studied the transcriptome of the anterior midgut of third instar larvae raised on an imbalanced diet. We took advantage of the specificity of the phenotype to narrow down our analysis. Indeed, Lp rescues AA imbalance in a control background (*Figure 1*), but not in an enterocyte-specific GCN2 knock-down background (*Figure 6*). Moreover, although ATF4 is one of the downstream targets of GCN2, ATF4 knock-down in enterocytes does not prevent the rescue of AA imbalance (*Figure 6G*). Finally, rescue of AA imbalance depends on the presence of r/tRNA operons in Lp (*Figure 2* and *Figure 6H, I*). Therefore, we took advantage of this series of phenotypical results informing on the

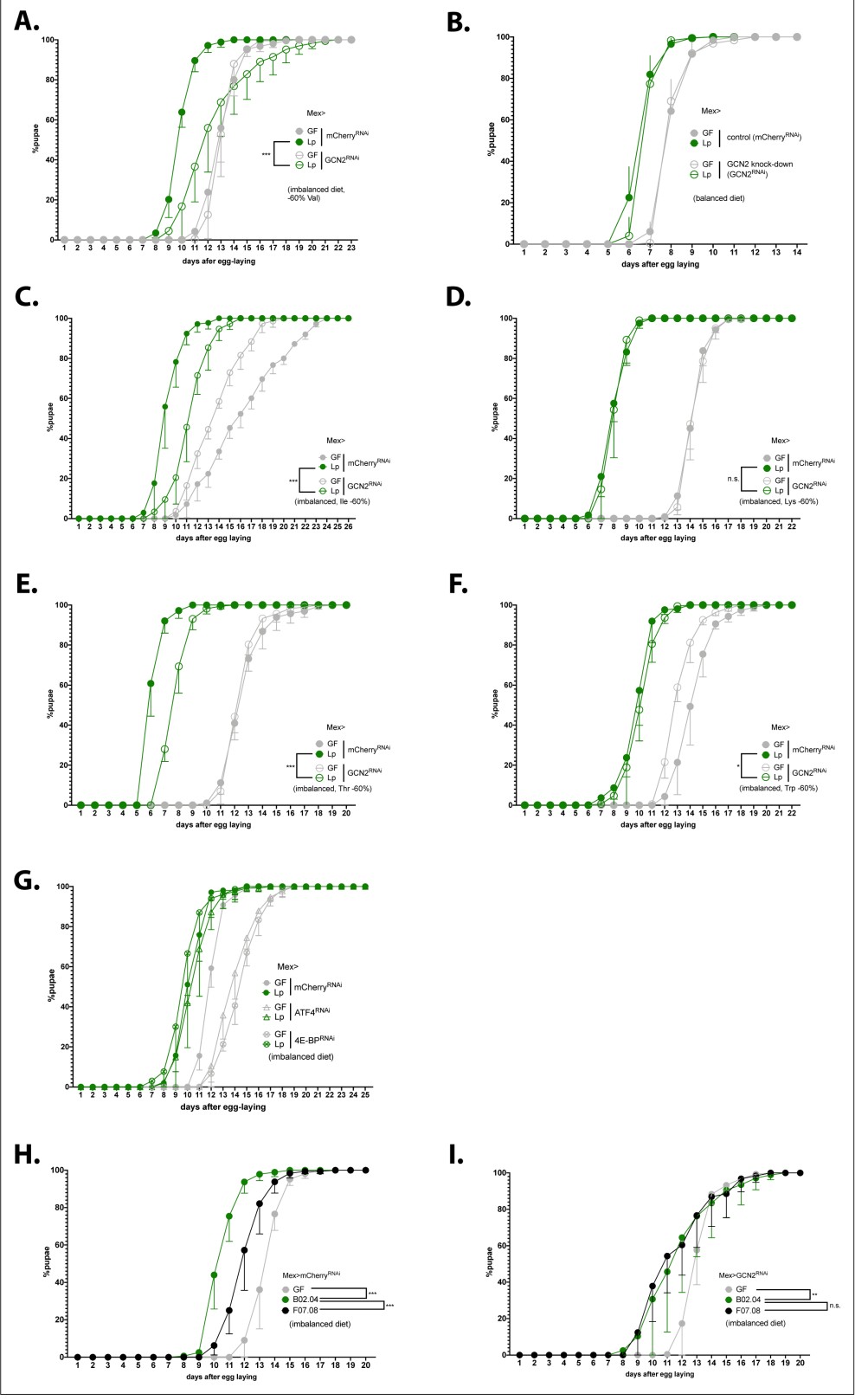

**Figure 6.** Expression of general control nonderepressible 2 (GCN2) in the gut is necessary for *L. plantarum* (Lp) to rescue the delay due to amino acid (AA) imbalance. (**A–I**) Developmental timing experiments. The graphs show five replicates per condition (mean and standard deviation). Each replicate consists in one tube containing 40 larvae. (**A–G**) Developmental timing of larvae raised in germ-free (GF) condition (grey circles) or Lp-associated

*Figure 6 continued on next page*

*Figure 6 continued*

conditions (green circles), in a control background (Mex>mCherry^RNAi, filled circles) or in a knock-down background in enterocytes (empty circles). The graphs represent the total fraction of emerged pupae over time as a percentage of the final number of pupae. When indicated, we used a Cox proportional hazards model to compare the effect of Lp in the control background and in the GCN2 knock-down background. n.s.: non-significant. **: p-value <0.01. ***: p-value <0.001. (**A–F**) GCN2 knock-down in the enterocytes on imbalanced diet (–60% Val) (**A**), balanced diet (**B**), or imbalanced diet: –60% Ile (**C**), –60% Lys (**D**), –60% Thr (**E**), –60% Trp (**F**). (**G**) ATF4 knock-down in the enterocytes (triangles) and 4E-BP knock-down in enterocytes (crossed circles) on imbalanced diet (–60% Val). (**H–I**) Developmental timing of GF larvae (grey), larvae associated with Lp:Tn_{r/tRNA} (black), and larvae associated with Lp:Tn_{control} (green) on imbalanced diet (–60% Val) in a control background (Mex>mCherry^RNAi) (**H**) or in a GCN2 knock-down in the enterocytes (Mex>GCN2^RNAi) (**I**). We used a Cox proportional hazards model to compare the effect of Lp:Tn_{control} association with Lp:Tn_{r/tRNA} association and GF condition. n.s.: non-significant, **: p-value <0.01, ***: p-value <0.001.

The online version of this article includes the following source data and figure supplement(s) for figure 6:

**Source data 1.** Raw data displayed in *Figure 6*.

**Figure supplement 1.** Expression of GCN2 in the gut, but not in the fat body nor expression of TOR is necessary for *L. plantarum* to promote growth on an AA-imbalanced diet.

**Figure supplement 1—source data 1.** Raw data displayed in *Figure 6—figure supplement 1*.

**Figure supplement 2.** Unlike *L. plantarum*, the symbiont *Acetobacter pomorum* does not activate GCN2 in the host's enterocytes.

**Figure supplement 2—source data 1.** Raw data displayed in *Figure 6—figure supplement 2*.

key mediators to compare six different biological contexts: (1) GF or (2) Lp-associated control larvae, (3) Lp-associated Mex>GCN2^RNAi larvae, (4) Lp-associated Mex>ATF4^RNAi larvae and control larvae associated with either (5) Lp:Tn_{r/tRNA} or (6) Lp:Tn_{control}. We then searched for transcriptomic signatures (i.e. genes differentially expressed [DE]) meeting the following criteria: (1) DE between Lp-associated control larvae and GF control larvae, (2) DE (in the same direction as 1) between Lp-associated control larvae and Lp-associated GCN2 knocked-down larvae, (3) NOT DE or with the inverse direction of expression as 1 between Lp-associated control larvae and Lp-associated ATF4 knocked-down larvae, (4) DE (in the same direction as 1) between Lp:Tn_{control}-associated larvae and Lp:Tn_{r/tRNA}-associated larvae (*Figure 7A*).

Principal component analysis showed that Lp-associated samples are more distant of the GF samples regardless of the genotype (*Figure 7—figure supplement 1A*). The microbial status is thus a major factor explaining transcriptomic variation. Moreover, among Lp-associated larvae, control larvae are more distant of ATF4 knock-down and GCN2 knock-down. This pattern was expected because ATF4 is downstream GCN2, and thus genes regulated by ATF4 should also be regulated by GCN2. However, ATF4 knocked-down samples and GCN2 knocked-down samples are clearly separated, which indicates that our dataset may allow us to identify signatures of GCN2-dependent ATF4-independent gene regulation (*Figure 7—figure supplement 1A*). Even though control larvae associated with Lp:Tn_{control} and Lp:Tn_{r/tRNA} were closer, most samples were separated from each other by PC2 (*Figure 7—figure supplement 1B*). We applied the four aforementioned selective filters (*Figure 7A*) to our dataset and found 104 genes consistently DE (25 up-regulated and 79 down-regulated) by Lp in a GCN2-dependent, ATF4-independent, and r/tRNA operon-dependent manner (*Figure 7B and C*). These genes are listed in *Supplementary file 2*.

We then performed a Gene Ontology (GO) enrichment analysis to search for biological functions enriched within the 79 down-regulated genes, in search for biological functions repressed upon Lp association in a GCN2-dependent, ATF4-independent, and r/tRNA operon-dependent manner. We found a significant enrichment in GO terms associated with mitochondrial respiration, resistance to oxidative stress and negative regulation of cell proliferation (*Figure 7D*). Proliferative cells such as cancer cells, growing yeasts, or embryonic cells tend to favour oxidative glycolysis and fermentation over mitochondrial respiration (in cancer cells, this phenomenon is known as the Warburg effect; *Warburg, 1956*). Indeed, although mitochondrial respiration is more effective at producing ATP, fermentation allows the synthesis of intermediates such as NADPH and ribose-5-phosphate that are necessary for anabolism and cell growth. Moreover, fermentation allows to produce ATP at a faster rate (*Lunt and Vander Heiden, 2011*). Therefore, the decrease in genes responsible for mitochondrial

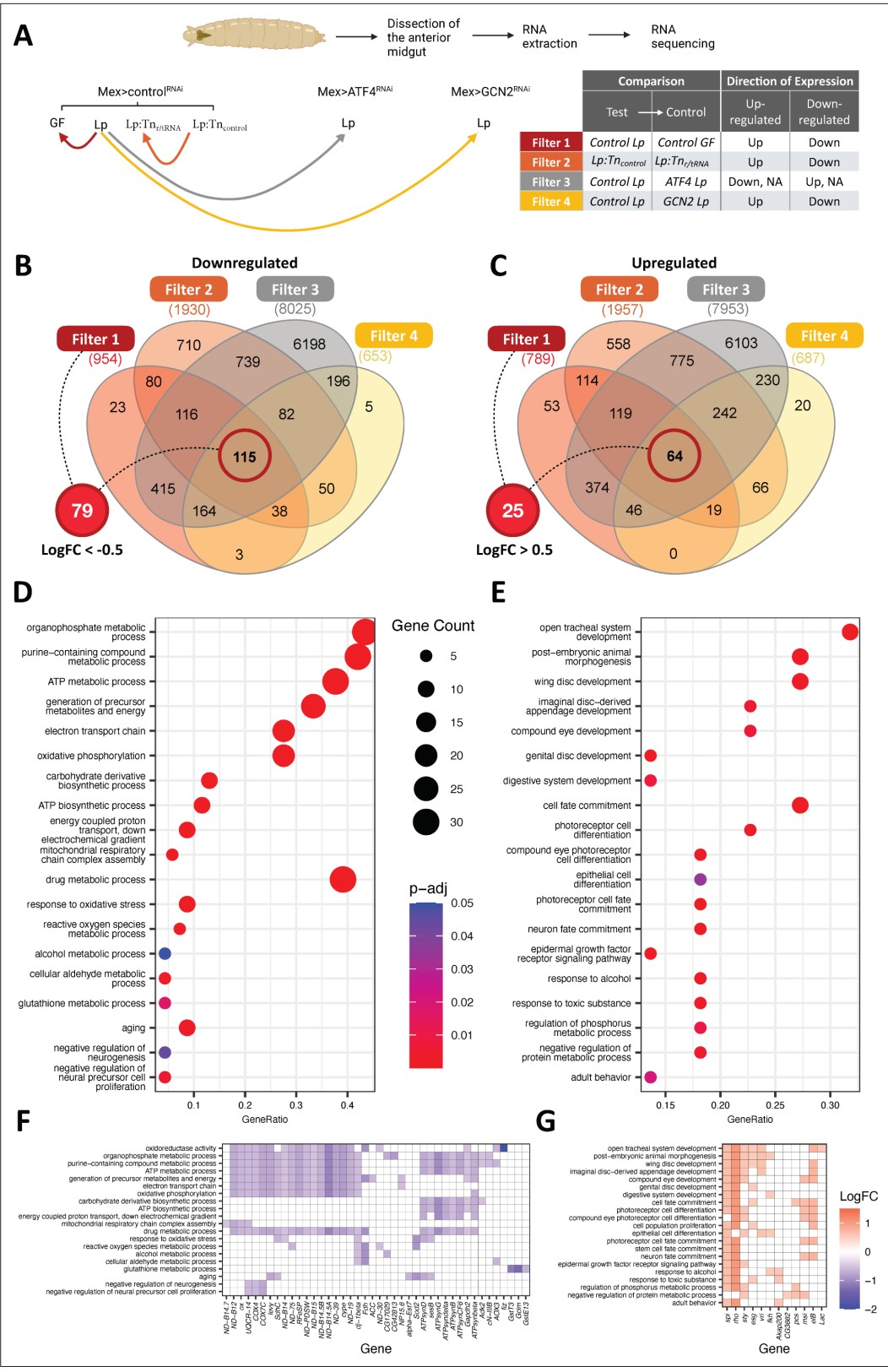

**Figure 7.** The anterior midgut gut of *L. plantarum* (Lp)-associated larvae display transcriptomic signatures of remodelling and metabolic switch. (**A**) Experimental design of the RNAseq strategy. We looked for genes differentially expressed (DE, red, orange, and yellow lines) or not differentially expressed (not DE, grey line) between conditions. Arrowheads indicate the control condition in each comparison. (**B, C**) Venn diagram showing

*Figure 7 continued on next page*

*Figure 7 continued*

the number of genes from each filter. We further selected genes from filter 1 with a Log2 fold change (LogFC) of less than –0.5 for genes down-regulated (**B**) and more than 0.5 for genes up-regulated (**C**) for functional enrichment analyses. (**D, E**) Dot plot of Gene Ontology (GO) enrichment analysis of genes down-regulated (**D**) or up-regulated (**E**) by Lp association in a general control nonderepressible 2 (GCN2)-dependent, ATF4-independent, and ribosomal and transfer RNA (r/tRNA) operon-dependent manner. The size of the dots represents the number of DE genes in each category, x-axis represents the gene ratio enrichment in comparison to the complete gene set of *D. melanogaster*; colour indicates the significance in terms of BH-adjusted p-values. (**F,G**) Corresponding LogFC plot from filter 1 (control Lp vs control germ-free [GF]) of genes belonging to the GO terms categories in (**D**) and (**E**), respectively.

The online version of this article includes the following source data and figure supplement(s) for figure 7:

**Source data 1.** Principal Component Analysis shows that association with bacteria and GCN2/ATF4 silencing have an important effect on the gut transcriptome.

**Figure supplement 1.** 3D principal component analysis (PCA) plot showing the first three components, calculated with a variance stabilising transformation of normalised counts from DESeq2.

respiration that we observe in the anterior midgut of Lp-associated larvae suggests that the organ is ongoing remodelling and proliferation. We then applied the same method to investigate the 25 up-regulated genes in Lp-associated larvae. We detected a significant enrichment for GO terms associated with organ morphogenesis, cell differentiation (different cell types: neurons or epithelial cells), and cell proliferation (especially members of the epidermal growth factor receptor [EGFR] pathway) (*Figure 7E*). These signatures suggest that Lp enhances the growth and maturation of the anterior midgut in a GCN2-dependent, ATF4-independent, and r/tRNA operon-dependent manner. Taken together, our transcriptomic analysis of the anterior midgut of larvae growing in an imbalanced diet allowed us to identify GCN2-dependent, ATF4-independent, and Lp r/tRNA operon-dependent signatures evoking increased epithelium maturation and altered metabolic activity which would together support tissue growth and maturation. We posit that this phenomenon would also support the systemic growth of the animal.

## Lp inhibits the expression of the systemic growth repressor *fezzik* in a GCN2-dependent manner

We noted that *fezzik* (*fiz*) was among the top genes whose expression was repressed by Lp in a GCN2-dependent, ATF4-independent, r/tRNA operon-dependent manner (*Figure 7F* and *Figure 8A*). *fezzik* is predicted to be an oxidoreductase (*Iida et al., 2007*), possibly involved in ecdysone metabolism (*McQuilton et al., 2012*). Importantly, larvae carrying an hypomorphic allele of *fezzik* or ubiquitously knocked down for *fezzik* display accelerated development (*Glaser-Schmitt and Parsch, 2018*), which suggests that *fezzik* is a systemic growth repressor. We knocked down *fezzik* specifically in enterocytes (*Figure 8—figure supplement 1*) and monitored the development of larvae on an imbalanced diet. *fezzik* knock-down in enterocytes accelerated the development of GF larvae, Lp-associated larvae, and LpΔop$_{r/tRNA}$-associated larvae (*Figure 8B*). These results suggest that *fezzik* inhibition through GCN2 activation and r/tRNAs contributes to the rescue of developmental delay by Lp on imbalanced diet.

## Discussion

Symbiotic microbes can interact with their host through symbiotic cues: molecules produced by the symbiont that functionally interact with the host signalling pathways and influence various aspects of its physiology. Here, we reveal a novel mechanism supporting the symbiosis between *Drosophila* and its bacterial partner Lp. We show that Lp can rescue the developmental delay of its host due to AA imbalance without providing the limiting AA. To understand how Lp promotes adaptation of its host to such dietary condition, we performed an unbiased genetic screen and showed that Lp operons encoding r/tRNAs are necessary to support host growth. We further showed that Lp activates the GCN2 signalling pathway in enterocytes of the anterior midgut. GCN2 activation depends on the presence of intact r/tRNAs-encoding operons in Lp and leads to a remodelling of the intestinal transcriptome suggesting tissue growth and maturation. GCN2 activation in the midgut is necessary to

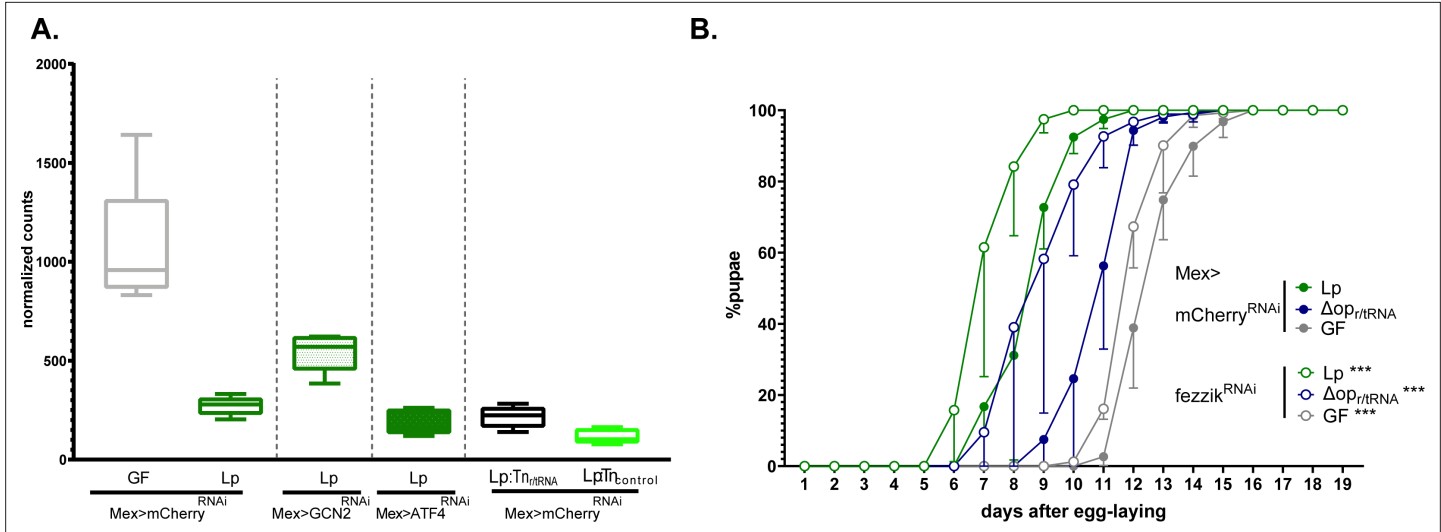

**Figure 8.** *L. plantarum* (Lp) inhibits the expression of the growth repressor *fezzik* in a general control nonderepressible 2 (GCN2)-dependent manner. (**A**) Expression of *fezzik* in germ-free (GF) larvae (grey) or larvae associated with Lp WT (dark green), Lp:Tn_r/tRNA (black), or Lp:Tn_control (light green) in a control background (Mex>mCherry^RNAi) or in larvae knocked down for GCN2 (Mex>GCN2^RNAi) or ATF4 (Mex>ATF4^RNAi) in enterocytes. The data is from the RNAseq described in *Figure 7*. (**B**) Developmental timing of GF larvae (grey) or larvae associated with Lp WT larvae (green) or LpΔop_r/tRNA in a control background (Mex>mCherry^RNAi, filled circles) or fezzik knock-down larvae (Mex>fezzik^RNAi, empty circles). The graph represents the total fraction of emerged pupae over time as a percentage of the final number of pupae. The graphs show five replicates per condition (mean and standard deviation). Each replicate consists in one tube containing 40 larvae. Larvae were reared on imbalanced diet (FLY AA –60% Val). We used a Cox proportional hazards model to compare the effect of fezzik knock-down in Lp, LpΔop_r/tRNA, and GF larvae. For each microbial condition, we used a post hoc Tukey test to compare the fezzik knock-down to control. ***: p-value <0.001.

The online version of this article includes the following source data and figure supplement(s) for figure 8:

**Source data 1.** Unlike *L. plantarum*, the symbiont *Acetobacter pomorum* does not activate GCN2 in enterocytes.

**Source data 2.** RT-qPCR showing the effect of fizz silencing by RNAi.

**Figure supplement 1.** *fezzik* RNAi efficiently represses the expression of *fezzik* in enterocytes.

support growth on an imbalanced diet, but not on a balanced diet or when symbiotic bacteria can synthesise the limiting AA (e.g. Lp in the context of lysine scarcity, or Ap upon valine scarcity). These results therefore reveal a mechanism by which a symbiont supports its host physiological adaptation to a suboptimal nutritional environment by engaging the host kinase GCN2 in enterocytes, a previously underappreciated function of this signalling module in the host gut.

## How is GCN2 signalling engaged by Lp?

We showed that GCN2 activation by Lp depends on the presence of the r/tRNA operons. GCN2 is known to be activated by several types of RNAs forming secondary and tertiary structures: eukaryotic tRNAs cognate to several AA (*Dong et al., 2000*), free ribosomal eukaryotic RNAs (*Zhu and Wek, 1998*), and viral RNAs (*Berlanga et al., 2006*). Our results show that purified bacterial tRNAs activate GCN2, which make Lp's tRNAs interesting candidates. Of note, one of the mutants (D12.16) shows disruption of an operon that encodes only rRNAs. However, synthesis of tRNAs may be regulated by ribosomal activity (*Gourse et al., 1985*); therefore, tRNAs production may be indirectly altered in D12.16 as well. Further work is needed to assess how the mutations that we identified impact the synthesis of each family of r/tRNA and to determine which of rRNA, tRNA or an indirect effect of the r/tRNA operon (for instance, through the translation rate in Lp that may be affected by r/tRNA mutations) is responsible for r/tRNA operon-dependent activation of GCN2 by Lp. Of note, the other two candidates that we identified through the genetic screen might also be linked to r/tRNA production: C08.20 has a transposon insertion of the gene *gnd2* that encodes a phosphogluconate dehydrogenase of the pentose phosphate pathway (PPP). One product of the PPP is the 5-phosphoribosyl-α-1-pyrophosphate, which is a precursor for the biosynthesis of nucleotides (*Kilstrup et al., 2005*). It is thus possible that disruption of *gnd2* in C08.20 might alter production of r/tRNAs by the cells. B12.11

displays an insertion in the end of an operon encoding *rsmC*, *lp_0696* and *lp_sRNA01*. *rsmC* encodes a 16S rRNA methyltransferase. Methylation of rRNA stabilises ribosomes and improves translation in other bacteria (*Wong et al., 2013*). *lp_0696* encodes a cytidine/deoxycytidylate deaminase, which catalyses conversion of cytidine into uridine. *lp_sRBA01* encodes the signal recognition particle (SRP), a small ncRNA which addresses membrane proteins to the membrane during their translation (*Kuhn et al., 2017*). Disruption of this operon may thus directly alter RNA production and/or ribosomal function, which can negatively regulate r/tRNA synthesis (*Gourse et al., 1985*). Several studies have shown that sRNAs from bacteria can act as cues sensed by host cells. tRNAs loci-derived sRNAs from bacteria can regulate host gene expression in the symbiotic nodules of plants (*Ren et al., 2019*). The sRNA SsrA is secreted by the symbiont *V. fischeri* and fosters immune tolerance of its host, the squid *Euprymna scolopes* (*Moriano-Gutierrez et al., 2020*). *P. aeruginosa* secretes tRNA-derived sRNAs that inhibit the immune response in cultured human cells and in the mouse lung (*Koeppen et al., 2016*). Here, we show that Lp produces extracellular vesicles containing mostly tRNAs. Extracellular vesicles can mediate host-microbes communications by transporting bacterial molecules (*Ñahui Palomino et al., 2021*). Therefore, we propose a model whereby Lp would deliver r/tRNAs to *Drosophila*'s enterocytes via extracellular vesicles. This model would explain why providing purified tRNAs directly to the larva does not fully recapitulate the effect of Lp on GCN2 activation, as if a vehicle (i.e. the extracellular vesicles) was needed. However, our model remains hypothetical: it is possible that r/tRNA production by Lp activates GCN2 through an undirect unidentified mechanism. We sequenced Lp's sRNA and we found that Lp's r/tRNA operons do not encode any additional sRNA; however, we observed that the expression of certain sRNAs was affected in LpΔop$_{r/tRNA}$. In particular, LpΔop$_{r/tRNA}$ shows down-regulation of SRP, which we identified in the genetic screen (see above) and tmRNA, which is the homolog of SsrA in Lp (*Moriano-Gutierrez et al., 2020*; *Supplementary file 1*). Further work may study whether these ncRNAs play a role in the symbiotic relationship of *Drosophila* and Lp.

We showed that Lp activates GCN2 signalling in enterocytes of the anterior midgut. This region of the midgut is of particular interest because it is located anteriorly to the acidic zone that kills a majority of Lp cells upon transit. It is thus the main region where enterocytes can interact with live Lp cells (*Storelli et al., 2018*). The pattern of expression of the 4E-BP$^{intron}$dsRed reporter slightly differs from the pattern observed by *Kang et al., 2017*: upon AA scarcity, they observed activation of the reporter in the gastric caeca, the acidic region and the proventriculus like we did, but not in the anterior midgut. Those experiments were done in conventionally reared larvae where the microbial status of the animals was not reported. We showed that GCN2 activation in the anterior midgut is symbiont-dependent; this difference with our findings thus suggests that the larvae in Kang and colleagues' experiments were associated with bacteria that do not promote GCN2 activation in the anterior midgut, such as Ap. An alternative explanation is that frequently flipped conventional fly stocks or crosses may carry very low microbial titres and therefore present GF-like phenotypes. Of note, we could still observe expression of the 4E-BP$^{intron}$dsRed reporter in the gastric caeca, the acidic region and the proventriculus upon GCN2 knock-down (whereas GCN2 knock-down completely abolishes 4E-BP$^{intron}$dsRed expression in the anterior midgut, *Figure 4C–F*). This suggests that other ATF4 activators such as PERK are active in the gastric caeca, the acidic region and the proventriculus. Importantly, we observed activation of GCN2 by Lp in both contexts of imbalanced and balanced diet (*Figure 4A and B*). We thus identified a GCN2 signalling activation mechanism which is AA-independent but bacteria-dependent. This differs from the canonical GCN2 activation relying on sensing unloaded eukaryotic tRNA upon AA scarcity. Previous studies have observed GCN2 activation in the *Drosophila* larval gut upon starvation (*Kang et al., 2017*), upon lack of EAA in adults (*Kim et al., 2021*), and upon exposure to a high sugar/low protein diet in adults (*Bonfini et al., 2021*). These observations were made without control over the fly's microbiota, and it is thus difficult to conclude whether bacteria may be at play in these contexts (especially because microbial composition depends largely on diet composition; *Lesperance and Broderick, 2020*). Studies have reported activation of GCN2 by pathogenic microbes: infection stimulates GCN2 in the gut, which triggers a translational block (*Chakrabarti et al., 2012*) and promotes immune response through the ATF4-4E-BP axis (*Vasudevan et al., 2017*). The mechanism of activation of GCN2 by infection was not identified in these studies. Of note, *Shigella* infection causes GCN2 activation in HeLa cells, and it was proposed that GCN2 is activated by AA depletion caused by the intracellular infection (*Tattoli et al., 2012*). However, this mechanism seems unlikely in our situation because Lp is not intracellular, and because

association with Lp actually promotes physiological rescue of AA scarcity. In the light of our results, we suggest that GCN2 may also be activated by r/tRNA produced by pathogenic bacteria upon infection. These studies and ours emphasise the role of GCN2 in the sensing of either pathogenic or symbiotic bacteria, in addition to its role in sensing uncharged eukaryotic tRNAs (*Donnelly et al., 2013*) and raise the question of the evolutionary history of GCN2 function: could GCN2 have primarily evolved as a pattern recognition receptor, that is a sensor for bacterial RNA later co-opted to sense AA scarcity indirectly through sensing uncharged eukaryotic tRNAs?

## How does GCN2 activation in the gut support improved systemic growth despite imbalanced nutrition?

Our work shows that GCN2 expression in enterocytes is necessary for Lp's support to larval growth on an AA-imbalanced diet. However, overexpression of a constitutively active form of GCN2 in dopaminergic neurons represses growth on an AA-imbalanced diet by inhibiting food intake (*Bjordal et al., 2014*). These apparently contradictory results suggest that the action of GCN2 is organ-specific: in dopaminergic neurons, GCN2 may foster adaptation to an imbalanced diet by prompting the larva to find a better food source; whereas in the midgut, GCN2 may allow adaptation to an imbalanced diet through physiological changes of enterocytes.

Studies in yeast's or *Drosophila*'s GCN2 pathway have focused on the effector ATF4, which translation increases upon eIF2 phosphorylation by GCN2. Especially, Vasudevan and colleagues showed than the GCN2-ATF4-4E-BP cascade yields the production of AMPs that help *Drosophila* fight infection (*Vasudevan et al., 2017*). Here, although we did indirectly observe activation of ATF4 (*Figure 4*) and 4E-BP (*Figure 4—figure supplement 1C*) upon Lp-association, this activation was not necessary for Lp to rescue the effects of AA imbalance (*Figure 6G*). Adaptation to AA imbalance may occur independently of eIF2, through other substrates of GCN2. eIF2 is the only known substrate of GCN2 so far; however, Dang Do and colleagues showed that in the mouse liver, GCN2 does not regulate the same set of genes as PERK, another eIF2-kinase. This suggests that additionally to their common substrate eIF2, GCN2 and PERK have distinct substrates (*Dang Do et al., 2009*). eIF2-independent action of GCN2 was described in response to UV exposure (*Grallert and Boye, 2007*) and viral infection (*Krishnamoorthy et al., 2008*). Alternatively, support to growth despite AA imbalance by Lp may rely on eIF2 targets other than ATF4. ATF4-independent regulation of gene expression by eIF2 was described in *Drosophila* (*Malzer et al., 2018*), mice (*Guo and Cavener, 2007*), and cultured mammalian cells (*Harding et al., 2003*; *Wek and Cavener, 2007*). Interestingly, GCN2, but not ATF4, is necessary for adaptation of mice to a Leucine-depleted diet (*Zhang et al., 2002*). We took advantage of the specificity of the reported phenotype to find out how Lp may influence the physiology of the anterior midgut in a GCN2-dependent, ATF4-independent, and r/tRNA-dependent manner. Our transcriptomics data suggest that the anterior midgut of Lp-associated larvae shows increased tissue growth and maturation and reduced mitochondrial respiration. GO terms associated with cell differentiation, cell proliferation (such as EGFR signalling), and morphogenesis are enriched in the Lp-associated conditions. This is in line with previous transcriptomic analyses in the midgut of adult flies, which showed that EGFR signalling and cell proliferation are stimulated by the gut microbiota (*Broderick et al., 2014*). In addition, expression of the growth inhibitor *fezzik* is repressed in the Lp-associated conditions. We showed that inhibition of *fezzik* specifically in the midgut is sufficient to accelerate larval growth. Finally, genes related to mitochondrial respiration are more expressed in the GF condition when compared to Lp-associated larvae. This is suggestive of a metabolic switch from respiration to fermentation upon GCN2 activation by Lp that may favour host anabolism to increase the incorporation of cellular building blocks (such as AA, lipids, and nucleotides) into the host tissue (*Lunt and Vander Heiden, 2011*). Of note, GCN2 activation can trigger a metabolic switch from respiration to fermentation in mammals (*Longchamp et al., 2018*). Importantly, transcriptomic remodelling upon GCN2 activation by Lp seems to be dependent on the presence of the r/tRNA in Lp and independent of the expression of ATF4 in enterocytes. These signatures may help explain the phenotype of rescue of AA imbalance by Lp. In adults, midgut remodelling during reproduction promotes nutrient absorption and allows optimal fecundity (*Ahmed et al., 2020*; *Reiff et al., 2015*). We propose that a similar mechanism occurs in larvae under Lp association, through GCN2 activation dependent on the r/tRNA operon. Midgut remodelling via anabolism and maturation would improve nutrient absorption, including valine, correcting dietary valine scarcity and promoting systemic growth.

Wild *D. melanogaster* larvae exclusively grow on decaying fruits, which are processed by microbes such as *Lactobacilli* (*Flatt, 2020*); therefore, we speculate that *Drosophila* has adapted its nutrition to the presence of bacteria and optimised its capacity to benefit from essential nutrients produced by its symbiotic microbes. GCN2 may have evolved as a sensor of symbiotic cues, allowing the larva to couple its growth with the abundance of microbes present in the environment: when microbes are abundant, symbiotic cues may activate GCN2 in the gut, triggering gut growth, optimal nutrient absorption, and rapid organismal growth.

In conclusion, we showed that the symbiotic bacterium Lp can activate GCN2 in the enterocytes, possibly through direct sensing of r/tRNAs as symbiotic cues. GCN2 activation in enterocytes decreases the transcription of the growth repressor *fezzik*, and triggers a metabolic switch that may improve AA absorption and rescue the effects of AA imbalance on larval growth. GCN2 is highly conserved across eukaryotes (*Donnelly et al., 2013*), and is important for mouse adaptation to an AA-imbalanced diet (*Anthony et al., 2004*; *Guo and Cavener, 2007*; *Laeger et al., 2016*; *Zhang et al., 2002*, p. 2). Our study therefore paves the way to testing whether the molecular dialogue between symbiotic bacteria and GCN2 described here is conserved in the animal kingdom beyond insects.

## Materials and methods
### *Drosophila* lines and breeding
*Drosophila* stocks were maintained at 25°C with 12:12 hr dark/light cycles on a yeast/cornmeal medium composed of 50 g·L$^{-1}$ of inactivated yeast, 80 g·L$^{-1}$ of cornmeal, 4 mL·L$^{-1}$ of propionic acid, 5.2 g·L$^{-1}$ of nipagin, and 7.14 g·L$^{-1}$ of agar. All experiments were conducted in gnotobiotic flies derived from GF stocks. GF stocks were established as previously described (*Erkosar et al., 2014*) and maintained on yeast/cornmeal medium supplemented with antibiotics (50 µg·mL$^{-1}$ of kanamycin, 50 µg·mL$^{-1}$ of ampicilin, 10 µg·mL$^{-1}$ of tetracyclin, and 5 µg·mL$^{-1}$ of erythromycin). We verified axenicity by grinding GF flies using a Precellys 24 tissue homogeniser (Bertin Technologies, Montigny-le-Bretonneux, France) and plating the lysate on Man-Rogosa-Sharp (MRS) Agar (Carl Roth, Karlsruhe, Germany) and LB Agar (Carl Roth, Karlsruhe, Germany). We used *yw* flies (BDSC #1495) as a reference strain. The following lines were used: UAS-mCherry$^{RNAi}$ (BDSC #35785), UAS-TOR$^{RNAi}$ (BDSC #33951), UAS-GCN2$^{RNAi}$-1 (VRDC #103976 obtained from P. Leopold's lab; this line was used for all GCN2 knock-downs unless specified otherwise), UAS-GCN2$^{RNAi}$-2 (BDSC #35355), UAS-GCN2$^{RNAi}$-3 (BDSC 67215), UAS-ATF4$^{RNAi}$ (VDRC #109014), UAS-4E-BP$^{RNAi}$ (VDRC #36667), 4E-BP$^{del}$-1 (BDSC #9558), 4E-BP$^{del}$-2 (BDSC #9559), UAS-fezzik$^{RNAi}$ (VDRC #107089), 4E-BP$^{intron}$dsRed (obtained from H.D. Ryoo's lab), Mex1-Gal4 and C564-Gal4 from out stocks. We generated the line 4E-BP$^{intron}$dsRed, UAS-GCN2$^{RNAi}$ by recombining the lines 4E-BP$^{intron}$dsRed and UAS-GCN2$^{RNAi}$-1. For RNA sequencing, we used UAS-GCN2$^{RNAi}$-1 and UAS-ATF4$^{RNAi}$, which are both KK lines from VDRC. In order to reduce transcriptional noise, we thus used the KK line VDRC-60100 crossed to Mex1-Gal4 as the control condition.

### Holidic diets
The HDs were prepared following the protocol of *Piper, 2017*, at a total AA concentration of 10.7 g·L$^{-1}$. We made two changes to Piper and colleagues' protocol: we used a lower concentration of sucrose (5 g·L$^{-1}$) because we noted that this concentration is the best for GF larvae: higher sucrose concentrations are toxic and slightly delay development of GF larvae (data not shown). Moreover, we omitted the conservatives (propionic acid or nipagin) because they can alter bacterial growth. We worked in sterile conditions: tubes and egg-laying cages were UV-treated or autoclaved, and solutions were either autoclaved (first part containing agar, Leu, Ile, Tyr, sucrose, cholesterol, and traces, as well as the acetate buffer solution) or filter-sterilised (stock solutions of EAA, NEAA, Glu, Cys, vitamins, nucleic acids, and lipids precursors, folate). For all experiments involving transposon mutants, we supplemented the HD with erythromycin (5 µg·mL$^{-1}$). HD was stored at 4°C for maximum 1 week before use.

### Bacteria and culture conditions
We used the strain Lp$^{NC8}$ of LP and the strain Ap$^{WJL}$ of AP. Of note, *Lactiplantibacillus plantarum* is the new name of the species formerly known as *Lactobacillus plantarum* (*Zheng et al., 2020*). Conversely to other strains of LP, Lp$^{NC8}$ was not isolated from a fly but from grass silage (*Axelsson et al., 2012*);

we used it because it is as growth promoting as fly isolates and it can be efficiently targeted for genetic modifications (*Matos et al., 2017*). Lp$^{NC8}$ was grown overnight at 37°C without agitation in MRS Broth Medium (Carl Roth, Karlsruhe, Germany). The Lp mutant library was generated by random transposon insertion from Lp$^{NC8}$ (*Matos et al., 2017*). All Lp transposon insertion mutants were grown for 24 hr in MRS supplemented with erythromycin at 5 µg·mL$^{-1}$. The Lp$^{GFP}$ strain was generated from Lp$^{WJL}$ (*Ryu et al., 2008*; *Storelli et al., 2018*). Lp$^{GFP}$ was grown for 24 hr in MRS supplemented with chloramphenicol at 10 µg·mL$^{-1}$. Ap$^{WJL}$ was isolated from a fly's intestine (*Ryu et al., 2008*). Ap$^{WJL}$ was grown for 24 hr in Mannitol Broth composed of 3 g·L$^{-1}$ of Bacto peptone (Becton Dickinson), 5 g·L$^{-1}$ of yeast extract (Becton Dickinson), and 25 g·L$^{-1}$ of D-mannitol (Carl Roth) in a flask at 30°C under 180 rpm agitation.

## Developmental timing experiments

GF flies were placed in a sterile breeding cage overnight to lay eggs on a dish of HD similar to the HD used for the experiment. At d0, we collected the eggs and placed them in the tubes containing the HD. Unless stated otherwise, each experimental condition consisted in five tubes each containing 40 eggs. Eggs were then inoculated with 100 µL of sterile PBS 1× (GF condition) or with 100 µL of a culture of bacteria resuspended in PBS 1× (yielding ~2 × 10$^8$ CFUs of Lp and ~10$^7$ CFUs of Ap per tube). For the HK condition, the resuspension of Lp in PBS was incubated 3 hr at 65°C. After inoculation, the larvae were kept at 25°C with 12:12 hr dark/light cycles. The number of newly emerged pupae was scored every day until all pupae have emerged. The data are represented either as curves of pupariation over time or as the median time of pupariation (D50) calculated using a custom script on RStudio. We used the R package *Coxme* (*Therneau and Grambsch, 2000*) to make a Cox proportional hazard model adapted to curves of pupariation over time, replacing the event 'death' with the event 'pupariation' (*Rodrigues et al., 2021*). Each experiment was performed in five independent replicates (one replicate consists in a tube containing 40 eggs as explained above). To compare the effect of Lp on development across different diets (e.g. balanced HD vs imbalanced HD), we tested whether the interaction between absence/presence of Lp and the diet was significant using the model *pupariation~absence/presence of Lp * diet + (1|Replicate)* (1|Replicate accounts for replicates as a random factor). When necessary, we used the FDR method to correct for multiple comparisons. To compare one treatment to other treatments (e.g. to compare the developmental timing of Lp:Tn$_{control}$-associated larvae with GF larvae and Lp:Tn$_{r/tRNA}$-associated larvae), we performed a general linear hypotheses test (glht) from the package *multcomp* (*Hothorn et al., 2008*) using Tukey post hoc pairwise comparisons.

## Measurement of valine in haemolymph

*yw* larvae were reared on HD as described for Developmental timing experiments. We collected them at pre-wandering mid-L3, 1 day before the emergence of the first pupae (typically D5 AEL for Lp-associated larvae on balanced diet, D6 AEL for Lp-associated larvae on imbalanced diet or GF larvae on balanced diet, D10 AEL for GF larvae on imbalanced diet). For each replicate, 10 larvae were washed in PBS and dried with tissue. We then placed them on a glass slide and tore them open with forceps. For each larva, we collected 1 µL of haemolymph using a pipet tip and pooled the 10 samples in 90 µL PBS. AA concentration was then determined by following the protocol in *Consuegra et al., 2020b*. Samples were diluted in miliQ water with a known quantity of norvaline used as the internal standard. Each sample was submitted to protein hydrolysis in sealed glass tubes with Teflon-lined screw caps (6N HCl, 115°C, for 22 hr). After air vacuum removal, tubes were purged with nitrogen. All samples were mixed with 50 µL of ultrapure water, buffered with borate at pH 10.2, and derivatised with ortho-phthalaldehyde using the Agilent 1313A autosampler at room temperature. AA analysis was performed by HPLC (Agilent 1100; Agilent Technologies, Massy, France) with a guard cartridge and a reverse phase C18 column (Zorbax Eclipse-AAA 3.5 µm, 150×4.6 mm; Agilent Technologies). Separation was carried out at 40°C, with a flow rate of 2 mL·min$^{-1}$, using 40 mM NaH$_2$PO$_4$ (eluent A [pH 7.8], adjusted with NaOH) as the polar phase and an acetonitrile/methanol/water mixture (45:45:10, vol/vol/vol) as the nonpolar phase (eluent B). A gradient was applied during chromatography, starting with 0% of B and increasing to 100% at the end. Detection was performed by a fluorescence detector set at 340 and 450 nm of excitation and emission wavelengths. For this quantification, norvaline was used as the internal standard, and the response factor of Val was determined using a 250 pmol·µL$^{-1}$

standard mix of AA. We use the software ChemStation for LC 3D Systems (Agilent Technologies) for analysis.

## Genetic screen

The genetic screen was performed in the same conditions as other Developmental timing experiments, but we used 20 eggs per condition in small tubes and one replicate per condition. Each condition consisted in the inoculation of one transposon insertion mutant. The screen was divided into four batches. For each batch, we calculated the D50 of each mutant and the associated z-score. We then pooled the z-scores from the four batches and selected the ones above a threshold of 2.5. The 32 candidates were re-tested in a Developmental timing experiment of five replicates. We compared the candidates to the WT-like transposon insertion mutant Lp:Tn$_{control}$ (B02.04) from the library (transposon inserted in an intergenic region downstream *dnaJ*).

## Mapping of insertions by whole genome sequencing

The transposons inserted in the mutant's genomes are not bar-coded. To map them, we extracted the genomic DNA of each candidate using a kit UltraClean Microbial DNA isolation (MoBio, Jefferson City, MO, USA). Samples were quality-checked using a Qubit 4.0 HS DNA. To sequence genomic bacterial DNA, libraries were built using the Nextera DNA Flex Library Prep (Illumina, San Diego, CA, USA) starting from 500 ng of DNA (except for two samples for which 350 and 280 ng were used) and following the provider's recommendations. The 17 dual-indexed libraries were pooled in an equimolar manner and sequenced on a paired-end mode (2×75 bp) using a NextSeq500 Illumina sequencer and a mid-output run. More than 155 M of reads were obtained for the run generating between 7 M to 12 M of reads by sample. Data were analysed using Galaxy (*Afgan et al., 2016*). Briefly, for each mutant, we filtered all pairs of reads which had one of the two reads mapped on the transposon sequence. We gathered the paired reads and mapped them on the genome of Lp$^{NC8}$ (*Axelsson et al., 2012*) to identify the region in contact of the transposon. The genome of Lp$^{NC8}$ contains five operons encoding r/tRNAs that share high sequence similarities. Therefore, sequencing did not allow us to identify in which operon the insertion took place. We thus used operon-specific PCR to identify in which operon the transposon was inserted for each mutant. For each mutant, we used two primers specific of the transposon (OLB215 and OLB 221) and one primer specific of each r/tRNA operon (op1, op2, op3, op4, and op5). The sequences of all primers used in this study can be found in *Supplementary file 3*.

## Construction of LpΔop$_{r/tRNA}$

We deleted the operon encoding rRNA 5S, 16S, and 23S as well as tRNAs for Ala, Ile, Asn, and Thr that was independently identified in two insertion mutants from the genetic screen (C09.09 and F07.08/Lp:Tn$_{r/tRNA}$, see *Figure 2G*). We used homology-based recombination with double-crossing over as described in *Matos et al., 2017*. Briefly, Lp$^{NC8}$'s chromosomic DNA was purified with a kit UltraClean Microbial DNA isolation (MoBio, Jefferson City, MO, USA). The regions upstream and downstream of the operon were then amplified using Q5 High-Fidelity 2× Master Mix (New England Biolabs, Ipswich, MA, USA) with the following primers that contain overlapping regions with the plasmid pG+host9 (*Maguin et al., 1996*) to allow for Gibson Assembly: tRNAop_1, tRNAop_2, tRNAop_3, and tRNAop_4 (*Supplementary file 3*). The PCR fragments were inserted into the plasmid pG+host9 digested with PstI (New England Biolabs, Ipswich, MA, USA) by Gibson Assembly (New England Biolabs, Ipswich, MA, USA) and transformed into *E. coli* TG1. The plasmid was purified and transformed into *E. coli* GM1674 *dam⁻dcm⁻* (*Palmer and Marinus, 1994*) to demethylate the DNA. The plasmid was then purified and transformed into Lp$^{NC8}$ by electroporation. Transformants were grown in MRS supplemented with erythromycin at 5 µg·mL⁻¹ at 42°C, which does not allow replication of the plasmid, in order to select integration of the plasmid into the chromosome by crossing-over. Plasmid excision associated with deletion of the operon by a second crossing-over event was then obtained by sub-culturing Lp in absence of erythromycin. The deletion was screened by colony PCR and confirmed by sequencing using the primers tRNAop_5 and tRNAop_6 (*Supplementary file 3*).

## Microscopy

4E-BP$^{intron}$dsRed larvae were reared on HD as described for Developmental timing experiments. We collected them at pre-wandering mid-L3, 1 day before the emergence of the first pupae (typically D5

AEL for Lp-associated larvae on balanced diet, D6 AEL for Lp-associated larvae on imbalanced diet or GF larvae on balanced diet, D7 AEL for GCN2 knocked-down Lp-associated larvae on imbalanced diet, D10 AEL for GF larvae on imbalanced diet, D6 AEL for Lp:Tn$_{control}$-associated larvae on imbalanced diet and D8 AEL for Lp:Tn$_{r/tRNA}$-associated larvae on imbalanced diet). For short-term association, GF larvae were reared on imbalanced diet until D8 AEL, associated with Lp:Tn$_{control}$ and Lp:Tn$_{r/tRNA}$ as previously described and collected at D10 AEL. Larvae were dissected in PBS 1×. The guts were fixed in paraformaldehyde (PFA) 4% in PBS 1× 1 hr at room temperature, washed in PBS 1×, washed three times in PBS Triton 0.2%, washed in PBS1X and mounted in ROTIMount FluorCare DAPI (Carl Roth, Karlsruhe, Germany). Pictures were acquired with a confocal microscope Zeiss LSM 780 (Zeiss, Oberkochen, Germany). We analysed the images on Fiji (*Schindelin et al., 2012*) using a custom macro: the macro identifies the reporter-positive regions above a defined threshold in the anterior midgut, and count the number of DAPI-positive particles inside these regions. It then counts the total number of DAPI-positive particles in the anterior midgut and uses it for normalisation.

## Bacterial growth on HD

Microtubes containing 400 µL of imbalanced HD were inoculated with ~10$^6$ CFUs of each candidate mutant. Five L1 GF larvae were added to each tube, and the tubes were incubated at 25°C. Each day, three samples per condition were collected for CFUs counting: we added 600 µL of sterile PBS 1× and grinded them using a Precellys 24 tissue homogeniser (Bertin Technologies, Montigny-le-Bretonneux, France. Settings: 6000 rpm, 2×30 s, 30 s pause). The homogenates were diluted at the appropriate concentration and plated on MRS Agar using an Easyspiral automatic plater (Interscience, Saint-Nom-la-Breteche, France). Plates were incubated at 37°C for 48 hr, and the number of CFUs was assessed using an automatic colony counter Scan1200 (Interscience, Saint-Nom-la-Breteche, France) and its counting software.

## Colonisation of the larval gut

Larvae were reared on imbalanced diet as for Developmental timing experiments. Six days AEL, larvae were collected, surface-sterilised in ethanol 70%, and grinded using a Precellys 24 tissue homogeniser (Bertin Technologies, Montigny-le-Bretonneux, France. Settings: 6000 rpm, 2×30 s, 30 s pause). The CFUs were then counted as described above.

## tRNAs feeding

4E-BP$^{intron}$dsRed larvae were reared on balanced diet as for Developmental timing experiments. At d0, d2, d4, and d5 AEL, the tubes were supplemented with 50 µL of a solution of tRNAs dissolved in Millipore water to reach a total concentration in the tube of 5, 25, and 125 µg·mL$^{-1}$. GF controls were supplemented with the same volume of Millipore water. We purchased the purified tRNAs at Sigma-Aldricht (St. Louis, MO, USA; bacterial tRNAs from *E. coli* 10109541001, eukaryotic tRNAs from yeast 10109517001). Larvae were dissected 6 days AEL and treated as described above.

## Food intake experiments

Larvae were reared on HD as described for Developmental timing experiments. Larvae were collected 1 day before the emergence of the first pupae and placed on HD containing Erioglaucine disodium salt (Sigma-Aldrich, St. Louis, MO, USA) at 0.8%. Every hour, we collected five larvae in five replicates per condition, rinsed them in PBS, and placed them in a microtube with beads and 500 µL PBS. Larvae were grinded using a Precellys 24 tissue homogeniser (Bertin Technologies, Montigny-le-Bretonneux, France. Settings: 6000 rpm, 2×30 s, 30 s pause). Optical density at 0.629 nm was measured using a spectrophotometer SPECTROstarNano (BMG Labtech GmbH, Ortenberg, Germany).

## RNA extraction of larval midgut

Larvae were reared as described for Developmental timing experiments and collected 1 day before the emergence of the first pupae. Larvae were dissected in PBS, and dissected anterior midguts were kept in RNAlater (Thermo Fisher, Waltham, MA, USA) before they were transferred to a microtube and flash-frozen. We used 10 guts for each replicate, and made five replicates for each condition. Samples were grinded using a Precellys 24 tissue homogeniser (Bertin Technologies, Montigny-le-Bretonneux,

France. Settings: 6500 rpm, 2×30 s, 30 s pause) and total RNA was extracted using a RNeasy kit (Macherey-Nagel, Hoerdt, France) following the instructions of the manufacturer.

## RNA extraction of Lp cultures

Lp was grown in liquid-imbalanced HD (Val –60%) for 5 days (*Consuegra et al., 2020b*). RNA was extracted following a procedure adapted from *Nakashima et al., 2020*. Briefly, cells were washed in PBS 1× and resuspended in 500 µL TRI reagent (QIAGEN, Hilden, Germany) with glass beads. Cells were vortexed for 3 min, incubated at 55°C for 30 min and centrifuged at 12,000 × *g* for 10 min at 4°C. We added 100 µL of chloroform, vortexed the tubes for three minutes, and centrifuged at 12,000 × *g* for 10 min at 4°C. We then collected the upper layer, added 160 µL of absolute ethanol, and applied it to a spin column of miRNeasy MiniKit (QIAGEN, Hilden, Germany). RNA was purified using the miRNeasy MiniKit following the instructions of the manufacturer.

## RT-qPCR

We adjusted RNA concentrations and performed reverse transcription (RT) on extracted RNAs using a SuperScript II RT kit (Thermo Fisher, Waltham, MA, USA) and random primers (Invitrogen, Waltham, MA, USA) following the instructions of the manufacturer. We then performed quantitative PCR using the primers GCN2-forward, GCN2-reverse, 4E-BP-forward, 4E-BP-reverse, rp49-forward, and rp49-reverse for *Drosophila* and 16S-forward, 16S-reverse, 23S-forward, 23S-reverse, Thr-tRNA-forward, Thr-tRNA-reverse, gyrB-forward, and gyrB-reverse for Lp (*Supplementary file 3*) using SYBR GreenER qPCR Supermix (Invitrogen, Waltham, MA, USA).

## RNA sequencing on larval anterior midgut

Thirty samples of total RNA isolated as previously described (five replicates per condition) were used to build libraries using the SENSE mRNA-Seq Library Prep Kit V2 from Lexogen and following the RTS protocol. The libraries were single-indexed and pooled together in an equimolar manner in order to sequence 20 libraries at a time on a high-output run in a single-end mode (1×86 bp) using a NextSeq500 Illumina sequencer. The two runs performed generated more than 535 M reads each, resulting in an average of around 26–27 M reads per sample. Statistics of sequencing can be found in *Supplementary file 4*.

## RNA sequencing analyses of the larval anterior midgut

We verified the quality of the samples using MultiQC on Galaxy (*Afgan et al., 2016*). Adapters and low-quality reads were removed with the help of Cutadapt on Galaxy (*Martin, 2011*). Clean reads were mapped using RNA STAR with default parameters (*Dobin et al., 2013*) against the genome of *D. melanogaster* (BDGP6.32.104). SARTools (*Varet et al., 2016*) was used for quality check of raw and normalised reads. Gene counts were normalised and tested for differential expression with DESEq2 package from R (*Love et al., 2014*). Genes were considered DE when the BH adjusted p-value (p-adj) was lower than 0.05. Common significantly expressed genes in multiple conditions were tested for GO enrichment with the clusterProfiler package from R (*Yu et al., 2012*) and non-redundant GO terms were reduced with the use of REVIGO (*Supek et al., 2011*).

## sRNA sequencing and processing

Lp WT and Lp Δop_{r/tRNA} were grown in triplicates in liquid-imbalanced HD (Val –60%) for 5 days (*Consuegra et al., 2020b*). RNA was extracted as described above. Libraries were built from 250 ng of total RNA using the NEBNext Multiplex Small RNA Library Prep Set for Illumina following the manufacturer's recommendations (New England Biolabs, Ipswich, MA, USA). Thirteen PCR cycles were completed to finalise the library preparation. The resulting indexed libraries were size-selected and purified using SPRI beads (Beckman-Coulter) with 1× ratio. Quantification and quality assessment of each library were performed with Qubit 4.0 (HS DNA kit, Thermo Fisher) and 4150 Tapestation analyser (D5000 ScreenTape kit, Agilent) respectively. Libraries were pooled in an equimolar manner and sequenced on an Illumina NextSeq 500 sequencer with HighOutput reagents. A total of 420 M sequence reads were obtained with 63 M to 73 M of reads per sample. Raw reads were processed using Cutadapt v1.18 (*Martin, 2011*) to remove adapters and short reads (length <14). The reads were mapped against the genome of Lp NC8 with Bowtie v1.13 (*Langmead et al., 2009*) with specific

parameters (*-n 0 -e 80 l 18 –best*) and were filtered for different annotated classes of rRNA, tRNA, ncRNAs, and coding regions. Sequencing reads were counted with featureCounts v2.0.1 (*Liao et al., 2014*). The complete set of ncRNAs along with gene counts were used to normalise read counts with the DESeq2 package v1.34.0 (*Love et al., 2014*). Selected tRNAs were also individually analysed for differential abundance in Prism GraphPad.

### Annotation of novel ncRNAs from Lp NC8

The complete genome of Lp NC8 (*Axelsson et al., 2012*) comprised a total of 60 tRNA loci, 15 rRNA loci, along with three ncRNAs. The complete genome sequence of Lp NC8 was reannotated in search of novel ncRNAs with prokka v1.13 (*Seemann, 2014*), and a comprehensive bacterial ncRNA database by combining BSRD (*Li et al., 2013*; *Nakashima et al., 2020*) and PRESRAT (*Kumar et al., 2021*). This final list of novel annotated ncRNAs in Lp NC8 (*Supplementary file 1*) was used in the subsequent mapping step.

### Figures and statistics

Figures were created using the Prism GraphPad software, Biorender (BioRender.com) and Rstudio packages clusterProfiler (*Yu et al., 2012*), ggplot2, and plotly. Statistical analyses were performed using the Prism GraphPad software and RStudio.

### Isolation of extracellular vesicles

We used a protocol adapted from *Li et al., 2017*. Supernatants from overnight cultures of Lp in MRS were generated by centrifuging cultures at 5000 × *g* for 10 min. Supernatants were then passed through a 0.22 µm filter to remove larger particles. Microvesicles were isolated using ExoQuick-TC kit (System Biosciences, Palo Alto, CA, USA) accordingly to the manufacturer's instructions. As a negative control, we applied the same procedure to sterile uninoculated MRS.

### Transmission electron microscopy

Isolated extracellular vesicles from Lp were placed onto a grid with formvar for 1 min after an hydrophilisation by etching process. The samples were contrasting with a 2% uranylacetate aqueous solution for 3 min. The mixture was washed in distilled water. Then, extracellular vesicles were observed with a TEM Jeol 1400 Flash and images were obtained with a Gatan camera ultrascan 1000.

### RNAse treatment, RNA extraction, and RT-qPCR from extracellular vesicles

For RNAse treatment, supernatants were treated with RNase 4 mg·mL$^{-1}$ (Sigma-Aldricht, St. Louis, MO, USA) for 10 min at 37°C before precipitation. RNase was inhibited by the addition of 1 µL of Murine RNase inhibitor (New England Biolabs, Ipswich, MA, USA) as described in *Moriano-Gutierrez et al., 2020*. RNA extraction and RT-qPCR was then performed on extracellular vesicles using the same protocol as for Lp cultures. We compared RNA levels from extracellular vesicles to the baseline consisting of sterile uninoculated MRS that went through the same procedures.

### Acknowledgements

We would like to thank Prof. Hyung Don Ryoo, Dr. Pierre Leopold, Dr. Nathalie Arquier, the Vienna *Drosophila* Resource Center and the Bloomington Stock Center for fly lines; Dr. David Duneau for his help with statistical analyses; Dr. Gilles Storelli for critical reading of the manuscript; Dr. Filipe de Vadder and Anne Lambert for their help during the revision process; Lucie Fallone and Meline Garcia, trainees of the Master Biosciences of ENS de Lyon for their help with experiments; the ArthroTools platform and the PLATIM platform of the SFR Biosciences (UAR3444/US8) for fly facility and microscopy equipments; the Institut de Biologie et Chimie des Protéines (IBCP) for their help with vesicles analysis; the Centre Technologique des Microstructures (Villeurbanne) for their help with the TEM imaging. Research in F Leulier's lab is supported by the 'Fondation pour la Recherche Médicale' ('Equipe FRM DEQ20180339196') and the Scientific Breakthrough Project from Université de Lyon 'Microbehave'. T Grenier was funded by a PhD fellowship from ENS de Lyon. J Consuegra was funded by a postdoctoral fellowship from the 'Fondation pour la Recherche Médicale' (FRM, SPF20170938612).

## Additional information

### Funding

| Funder | Grant reference number | Author |
|---|---|---|
| Fondation pour la Recherche Médicale | DEQ20180339196 | Théodore Grenier<br>Jessika Consuegra<br>Houssam Akherraz<br>Longwei Bai<br>Isabelle Rahioui<br>Pedro Da Silva<br>Benjamin Gillet<br>Sandrine Hughes<br>Cathy I Ramos<br>Renata C Matos<br>François Leulier |
| Fondation pour la Recherche Médicale | SPF20170938612 | Jessika Consuegra |

The funders had no role in study design, data collection and interpretation, or the decision to submit the work for publication.

### Author contributions

Théodore Grenier, Conceptualization, Formal analysis, Investigation, Methodology, Writing – original draft; Jessika Consuegra, Mariana G Ferrarini, Investigation, Methodology; Houssam Akherraz, Yves Dusabyinema, Isabelle Rahioui, Benjamin Gillet, Sandrine Hughes, Methodology; Longwei Bai, Pedro Da Silva, Investigation; Cathy I Ramos, Renata C Matos, Supervision, Investigation; François Leulier, Conceptualization, Supervision, Funding acquisition, Validation, Project administration, Writing - review and editing

### Author ORCIDs

Théodore Grenier [ID] http://orcid.org/0000-0001-6022-1954
Jessika Consuegra [ID] http://orcid.org/0000-0002-3185-803X
Renata C Matos [ID] http://orcid.org/0000-0001-7480-6099
François Leulier [ID] http://orcid.org/0000-0002-4542-3053

### Decision letter and Author response

Decision letter https://doi.org/10.7554/eLife.76584.sa1
Author response https://doi.org/10.7554/eLife.76584.sa2

---

## Additional files

### Supplementary files

• Supplementary file 1. Non-coding RNAs expressed by Lp grown in HD. L/s: sRNA (s) or leader sequence (L). Log2 Fold Change: expression in Lp WT vs expression in Lp $\Delta op_{r/tRNA}$.

• Supplementary file 2. Differentially expressed genes and Gene Ontology (GO) terms in larval anterior midgut RNAseq.

• Supplementary file 3. Primers used in this study.

• Supplementary file 4. Statistics for larval anterior midgut RNAseq.

• MDAR checklist

• Source code 1. Cox proportional hazard model applied to pupariation.

### Data availability

RNAseq raw data are deposited at NCBI Sequence Read Archive under the numbers SUB10970982 and PRJNA799161. Source data files containing all the numerical data used to generate the figures are provided for each figure.

The following dataset was generated:

| Author(s) | Year | Dataset title | Dataset URL | Database and Identifier |
|---|---|---|---|---|
| Grenier T, Galvao Ferrarini M, Akherraz H, Dusabyinema Y, Gillet B, Hughes S, Leulier F | 2023 | Bioproject PRJNA799161 | https://www.ncbi.nlm. nih.gov/sra/?term= PRJNA799161 | NCBI Sequence Read Archive, PRJNA799161 |

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
