## [Editor Report]

Previous studies found that a component of the microbiota, *Lactobacillus plantarum*, can provide support to its host *Drosophila melanogaster* during development. Here, the authors further explore this interaction using defined diets where they find that under conditions that have low levels of some essential amino acids, the bacteria can still promote survival even though the bacteria is not synthesizing the amino acid. Through a screen of bacterial transposon insertion mutants, these authors identify bacterial transfer and ribosomal RNAs as necessary for this effect, and studies in the fly demonstrate that the host kinase GCN2, a protein known to associate with host tRNAs, in enterocytes is the mediator of this response. This manuscript links the intestinal microbiota to host protective responses providing important insights into these interactions.

---

## [Decision Letter]

**Decision letter after peer review:**

Thank you for submitting your article "Intestinal GCN2 controls *Drosophila* systemic growth in response to *Lactiplantibacillus plantarum* symbiotic cues encoded by r/tRNA operons" for consideration by *eLife*. Your article has been reviewed by 3 peer reviewers, one of whom is a member of our Board of Reviewing Editors, and the evaluation has been overseen by Utpal Banerjee as the Senior Editor. The reviewers have opted to remain anonymous.

Essential revisions:

All three reviewers found the work to be interesting however, found that there was a lack of proof of RNA transfer to host cells. Why the mechanisms remain unclear, there are indeed other examples of such a phenomenon. For example, FISH (RNA and DNA) can be used to demonstrate bacterial RNA in host cells. Studies demonstrating this transfer are required to support the conclusions.

Another question raised regards how GCN2 allows the larva to correct or deal with the amino acid imbalance. Any additional insights here would also be helpful for the audience.

*Reviewer #1 (Recommendations for the authors):*

This is an interesting article that uses the power of *Drosophila* to explore how organisms work with their symbionts to adapt to a changing environment. The authors show that reducing some non-essential amino acids that cannot be produced by the "symbiont" *Lactobacillus* can nevertheless be rescued by the presence of this bacteria. Why not Leucine? What makes valine and isoleucine different? And how does the adaptation work? They suggest it is not through provisioning from the bacteria by using heat killed bacteria. Therefore, there should be an overall reduction in total Valine in the animal. Is this the case? And if so, what causes the delay that is overcome? Are there less proteins made (measure total valine in proteins from different diets). What is adaptation here?

To explore this interaction, they begin to genetically dissect the requirements for protection in the bacteria. Further characterization of the bacterial mutants would be useful here to show the delay over time for decreasing concentrations of Leucine and to determine if this is specific to leucine. Figure 2F done with altered % and other aa. Show time courses and not the bar graphs. They then show that the mutants have transposon insertions in r/tRNA loci and reduced rRNA levels. What is its protein concentration? Is this different in these bacteria? And are these RNAs released out of cells? What is their half life? And how would they cross membranes? Nucleic acids cannot passively cross. Can be measured in media from the different strains. The calculations of the amount of these species assumes all of the nucleic acid inside the bacteria is accessible to the inside of the enterocyte.

Experiments next demonstrate that colonization with Lp leads to induction of an ATF4 reporter independent of diet. But that colonization of the mutant Lp, has reduced activation during a balanced diet but not in an imbalanced diet. This was also the case for a mutant identified in the screen. I am not sure what this means since the phenotypes are during an imbalanced diet. Moreover, feeding flies tRNA can induce some reporter but reduced compared to even the operon deleted strain of Lp. Does this impact D50 or %pupation? Does it also induce reporter in an imbalanced diet? Need to show more data around the tRNA experiments. This is a major conclusion.

The experiments in figure 5 show that there are selective requirements for GNC2 depending on the diet and aa imbalance. This is very complicated. As the depletion of GCN2 in 5a does not impact GF pupation on an imbalanced diet, it does for other alleles in S5A-B. This is striking and impacts some of the conclusions. What does it look like for these alleles on a balanced diet? Moreover, the day to 50% pupation is variable (eg controls in A different than controls in G). Only one Lp mutant tested and the graphs are spit between H and I while shown altogether in A. Is there a difference between the WT bacteria in the two fly genotypes as seen in A? Interestingly, AP can rescue independent of GCN2. Thus there are different solutions to this problem. Moreover, the authors found that ATF4 and 4EBP are dispensable, so what is important about GCN2, is it eiF2a phosphorylation. Is translation impacted in enterocytes, is turnover? What is the mechanisms? Does feeding tRNAs impact GCN2 dependent pupation changes? Characterization of the gut cells, ISCs, turnover and physiology (eg metabolomics to monitor aa levels and incorporation) they perform RNAseq and describe a series of GO categories. No experiments either validate or explore the significance of these genes.

We are left with an incomplete understanding of how Lp impacts development, whether RNA is a signal and how this may be sensed to induce a survival program.

Experimental interpretations:

It is unclear how many independent experiments and how many animals per experiment are included in each figure. What do the aggregate values represent? How many of each? This should be described in detail for all experiments including % pupation experiments and confocal.

*Reviewer #2 (Recommendations for the authors):*

1) For ease of comprehension, it would be helpful to refer to the L. plantarum mutants by a descriptive designation such as the interrupted tRNA/rRNA or "control", in the case of the control mutant. It is difficult to remember the significance of the number designations.

2) Figure 4E and F: Here the authors feed GF *Drosophila* purified bacterial t-RNA's and then show that transcription of 4E-BP is increased in enterocytes. Feeding of an unrelated RNA of similar length would prove that tRNA, in particular, is required for this response.

3) Figure S4G: Feeding of both prokaryotic and eukaryotic tRNAs increases expression of 4E-BP. It would be helpful to show an alignment of the sequences of the eukaryotic and prokaryotic tRNAs used here and to comment on their similarity or lack thereof. Is the prokaryotic tRNA similar enough to eukaryotic tRNA to interact with GCN2 if it can be taken up?

4) Purified tRNAs have a much smaller effect on 4E-BP expression than L. plantarum. While one possibility is that a bacterial carrier such as OMVs are necessary for uptake into enterocytes, another possibility is that additional bacterial components are required for the response. Does a mixture of heat-killed bacteria and purified tRNAs yield a stronger response? While this experiment would require new reagents, the investigators could prove that transport is necessary by conditionally expressing a prokaryotic tRNA directly in enterocytes and showing that GCN2 is activated.

5) The data in Figure 5SE showing that L. plantarum does not alter food intake, might be better positioned earlier in the manuscript as this is a key control that does not rely on the data for tRNA or GCN2.

6) One of the interesting findings of this paper is that deceleration of development is not the direct result of the inability to access an essential amino acid but rather a regulated process that occurs when nutrition is suboptimal but actually adequate to support faster development. It is interesting to speculate on why this might be an adaptive response.

*Reviewer #3 (Recommendations for the authors):*

The work is essentially OK as is, except for a few specific issues outlined above.

The comments below aim to improve the impact of the article with suggestions for additional experiments.

1. Val absorption: would it be possible to monitor Val intake by the gut in pulse-chase experiments, using labelled valine which most likely would have to be radioactive?

2. Release of bacterial r/tRNAs: the authors mention the possibility of export through EVs. A simple experiment is to make an EV preparation and to check by PCR for the presence of r/tRNAs in the purified EVs. This experiment may also allow to answer the question as to whether the bacteria release the r/tRNAs in response to the imbalance of amino-acids in the culture medium (in other words, is the response to the imbalanced diet taking place at the level of the symbiont, the host, or both?). A more challenging one would be to feed the larvae with EVs from wt and mutant Lp bacteria. The difficulty may be to obtain enough material. Does a CAFE-like assay exist for larvae?

[Editors' note: further revisions were suggested prior to acceptance, as described below.]

Thank you for resubmitting your work entitled "Intestinal GCN2 controls *Drosophila* systemic growth in response to *Lactiplantibacillus plantarum* symbiotic cues encoded by r/tRNA operons" for further consideration by *eLife*. Your revised article has been evaluated by Utpal Banerjee (Senior Editor) and a Reviewing Editor.

The manuscript has been improved but there are some remaining issues that need to be addressed, as outlined below:

Please address the specificity questions raised about the vesicles by Reviewer 2.

*Reviewer #2 (Recommendations for the authors):*

The authors have responded to all the reviewers' comments in an impressively thorough way. While the effort to observe vesicles in the intestine is admirable, I agree with them that this is not convincing enough to include. I have only one concern regarding the qRT-PCR on membrane vesicles. I would like to see a negative control/ absent RNA that might demonstrate selectivity. The expected RNAs are there. Are these found in the same ratio as in the bacterium? If so, are there any RNAs that are found at different levels in the bacterium and the vesicle? If all RNAs are found in vesicles in the same ratios as in the bacterium, then how to prove that these are not contaminating RNAs? After all, these are some of the most abundant RNAs in the bacterium. I am also concerned that ribosomes may be co-purifying with OMVs. Another possibility might be to treat the membrane vesicles with RNase as RNA in vesicles is thought to be resistant to the action of such enzymes. Of course, I fully appreciate the challenge of proving that OMVs carry rRNAs and tRNAs and the best one may be able to do is provide supporting evidence.

*Reviewer #3 (Recommendations for the authors):*

The authors have made a reasonable attempt to address the issues raised by the reviewers and it appears that this article has now only to address minor issues. Of note, may-be using RNAScope probes might have given the level of sensitivity required to detect bacterial RNA within enterocytes.

---

## [Author Response]

Essential revisions:All three reviewers found the work to be interesting however, found that there was a lack of proof of RNA transfer to host cells. Why the mechanisms remain unclear, there are indeed other examples of such a phenomenon. For example, FISH (RNA and DNA) can be used to demonstrate bacterial RNA in host cells. Studies demonstrating this transfer are required to support the conclusions.Another question raised regards how GCN2 allows the larva to correct or deal with the amino acid imbalance. Any additional insights here would also be helpful for the audience.

We thank the editor and the referees for reviewing our work. Following the editor and referees’ suggestions, we performed additional experiments and made some changes to our manuscript which we feel address (at least partly) the major points raised by the reviewers.

In the discussion of our initial manuscript, we proposed a release of r/tRNA from *L. plantarum* (Lp) via extracellular vesicles. We have now directly tested this hypothesis and show the results in new Figure 3 of the updated manuscript. We now reveal that Lp produces extracellular vesicles that contain rRNAs and tRNAs, including those encoded by the r/tRNA operon under study. This observation indicates that extracellular vesicles may be one vehicle by which host cells acquire and sense symbiotic cues such as bacterial r/tRNAs.

Moreover, the editor suggested that we do FISH to visualize Lp’s r/tRNAs in host’s cells. We performed HCR-FISH on the gut of GF larvae and Lp-associated larvae using probes specific for Lp’s 16S rRNA (Akhtar et al., PLOS ONE 2021). In Lp-associated larvae, we detected large positive particles that may be extracellular vesicles (see Author response image 1 and Author response image 2). We did not see any signal in GF guts. Additionally, we did not detect any signal in a negative control consisting in performing the experiment with the hairpin probe only, without the secondary fluorescent probe, to account for potential background.

**Author response image 1. sa2fig1:** Representative images of the anterior midgut of larvae stained for Lp’s 16S rRNA (yellow) and DAPI (cyan). Scale bar: 50 µm.

The rRNA signal appears to be in the lumen, but we are unsure whether or not it is also in the enterocytes. The following pictures show positive dots in the gut of Lp-associated larvae that may be in the enterocytes.

**Author response image 2. sa2fig2:** Representative confocal picture of the anterior midgut of a Lp-associated larva. We selected a plane showing putative inclusions of Lp’s rRNA inside the enterocytes. Scale bar: 50 µm. Right picture: the red line depicts the border between the lumen (on the top, characterized by DAPI-positive bacteria) and the gut epithelium (on the bottom). The red arrows show putative inclusion of Lp’s 16S rRNA in the enterocytes.

In addition, please find Author response video 1 showing a z-stack of the anterior midgut of a Lp-associated larvae stained with HCR-FISH for Lp’s 16S rRNA (yellow) and DAPI (cyan).

**Author response video 1. sa2video1:** 

Although these images support the presence of Lp’s rRNA in host’s enterocytes, we do not consider that the level of evidence is robust enough for them to be included in the manuscript as we were only able to perform HCR-FISH in a few midguts and not in a systematic manner. We therefore propose to provide these data only in our response to the referees (which will be accessible along the published paper) and remain speculative about this point in our discussion (line 642-647).Furthermore, we now bring additional insights regarding the mechanisms by which GCN2 promotes larval growth upon AA imbalance. To this end, we capitalized on our functional transcriptomics approach reported in the initial manuscript. Among the genes that were down-regulated upon Lp association in a GCN2-dependant manner, we identified the growth repressor and putative Ecdysone-oxydase *fezzik* as an interesting candidate. Using RNAi, we now show that on its own the repression of *fezzik* expression in enterocytes of GF animals leads to improved growth, indicating that Lp-mediated GCN2-dependant *fezzik* downregulation contributes to larval growth upon Lp association. These data are provided in the new Figure 8 and the new Figure 8—figure supplement 1.

Moreover, we have now performed additional experiments to address several reviewers’ comments:

– We now show that Lp restores the Valine levels in the hemolymph of larvae deprived of Valine (new Figure 1E), which is in accordance to our hypothesis that Lp association improves gut functions and nutrient absorption, most likely by supporting gut maturation as indicated by our functional transcriptomics analysis.

– We now test the other GCN2-RNAi lines by RT-qPCR and showed that they efficiently knock-down GCN2 (new Figure 6—figure supplement 1D).

– We have now used two *Drosophila* lines carrying loss-of-function mutations in *thor* (*4E-BP*) to confirm our previous observation, based on RNAi, that *4E-BP* is not required for Lp to promote larval growth (new Figure 6—figure supplement 1I).

– We now show that the Lp deletion mutant for r/tRNA operon is impaired at promoting growth not only upon low Valine but also in the context of limiting Leucine (new Figure 2-supplement 1D), a result that reinforces the notion that Lp association improves gut functions and nutrient absorption, not just Valine uptake.

– Additionally, we wondered whether the r/tRNA operon may encode small non-coding (nc) RNA on top of ribosomal and transfer RNAs, as it has been described in other bacteria (Stenum et al., Frontiers in Microbiology 2021). We sequenced the ncRNAs expressed by Lp in culture and identified 13 ncRNAs expressed in these conditions. We now provide a list of these sncRNAs (new Supplementary file 1). None of them were encoded in the r/tRNA operon, which suggests that they do not play a role in the phenotype that we describe and reinforce the notion that r/tRNAs from the deleted locus are responsible for the phenotype observed. Moreover, we used this dataset to show that as expected, the expression of tRNAs is decreased in the Lp mutant for r/tRNA operon (new Figure 2-supplement 1I).

Finally, we made some corrections to the text and figures following the reviewers‘ recommendations.

We provide a point-by-point response to the reviewers’ comments below.

Reviewer #1 (Recommendations for the authors):This is an interesting article that uses the power of *Drosophila* to explore how organisms work with their symbionts to adapt to a changing environment. The authors show that reducing some non-essential amino acids that cannot be produced by the "symbiont" Lactobacillus can nevertheless be rescued by the presence of this bacteria. Why not Leucine? What makes valine and isoleucine different?

We thank the reviewer for his/her kind words. We focused only on essential amino-acids: when one of these amino-acids (Valine, Leucine, Isoleucine, Histidine…) is completely removed from the diet, the larvae cannot develop (Consuegra et al. 2020; Sang 1956). We did observe that decreasing Leucine concentration causes developmental delays in GF larvae, which is compensated by Lp (new Figure 1A-B). As the reviewers points out, Valine and Isoleucine seem to be “more essential” because reducing the amount of Valine or Isoleucine leads to a greater growth delay in GF larvae than reducing Leucine. We hypothesize that either Leucine is less important for larval growth, or the recipe of the holidic diet is biased toward an overabundance of Leucine.

And how does the adaptation work? They suggest it is not through provisioning from the bacteria by using heat killed bacteria. Therefore, there should be an overall reduction in total Valine in the animal. Is this the case? And if so, what causes the delay that is overcome? Are there less proteins made (measure total valine in proteins from different diets). What is adaptation here?

To answer the reviewer’s question, we measured the quantity of Valine in the hemolymph of size-matched larvae in different conditions (new Figure 1E). We observed a reduction in total Valine in GF larvae fed an imbalanced diet. This reduction was rescued by association with Lp, which supports our main hypothesis that Lp improves gut functions and nutrient absorption, including Valine.

To explore this interaction, they begin to genetically dissect the requirements for protection in the bacteria. Further characterization of the bacterial mutants would be useful here to show the delay over time for decreasing concentrations of Leucine and to determine if this is specific to leucine. Figure 2F done with altered % and other aa. Show time courses and not the bar graphs.

To further characterize the mutant, we assessed the time of development of larvae associated with Lp r/tRNA mutant (Lp Δop_r/tRNA_) on a diet with decreased Leucine concentration (-70%). We observed that similarly to what we observed upon Valine limitation, LpΔop_r/tRNA_ fail to promote larval growth upon Leucine limitation (new Figure 2-supplement 1D).

They then show that the mutants have transposon insertions in r/tRNA loci and reduced rRNA levels. What is its protein concentration? Is this different in these bacteria? And are these RNAs released out of cells? What is their half life? And how would they cross membranes? Nucleic acids cannot passively cross. Can be measured in media from the different strains. The calculations of the amount of these species assumes all of the nucleic acid inside the bacteria is accessible to the inside of the enterocyte.

We thank the reviewer for this interesting question. Upon passage through the intestine, the majority of Lp cells are killed when going through the acidic region (Storelli et al. 2018), which could cause a release of r/tRNAs. However, it is unknown whether free r/tRNA may be able to cross the membranes. Other studies have reported that extracellular vesicles produced by bacteria can contain bacterial RNAs. Host cells can uptake the content of extracellular vesicles (Brown et al. 2015). We now show that Lp produces extracellular vesicles and these vesicles contain r/tRNAs (new Figure 3). We therefore propose that Lp’s r/tRNAs can be transported into host’s cells through extracellular vesicles.

Experiments next demonstrate that colonization with Lp leads to induction of an ATF4 reporter independent of diet. But that colonization of the mutant Lp, has reduced activation during a balanced diet but not in an imbalanced diet. This was also the case for a mutant identified in the screen. I am not sure what this means since the phenotypes are during an imbalanced diet. Moreover, feeding flies tRNA can induce some reporter but reduced compared to even the operon deleted strain of Lp. Does this impact D50 or %pupation? Does it also induce reporter in an imbalanced diet? Need to show more data around the tRNA experiments. This is a major conclusion.

The reduction in ATF4 induction upon colonization by Lp Δop_r/tRNA_ is not significant (p-value=0.069), but we observed it also upon colonization by Lp:Tn_r/tRNA_ and upon short-term association (where the decrease reaches statistical significance, Figure 5-supplement 1E,F). This suggests that Lp Δop_r/tRNA_ leads to decreased GCN2 activation on both balanced and imbalanced diet. We tested whether purified tRNAs impacts developmental timing and found that it does not. This may suggest that low GCN2 activation in GF larvae is not sufficient to promote growth; however, as the reviewer points out the induction of the GCN2 reporter by purified tRNAs is very low. Especially, it is much lower than induction by Lp. Therefore, it is possible that purified tRNAs simply do not activate GCN2 enough to have a significant effect on growth.

The experiments in figure 5 show that there are selective requirements for GNC2 depending on the diet and aa imbalance. This is very complicated. As the depletion of GCN2 in 5a does not impact GF pupation on an imbalanced diet, it does for other alleles in S5A-B. This is striking and impacts some of the conclusions.

We measured the efficiency of the other two RNAi lines (GCN2-2 and GCN2-3) and found that they do knock-down GCN2 effectively, though the interference efficiency is reduced as compared to GCN2-RNAi-1 (new Figure 6—figure supplement 1D). Lp-associated larvae knocked-down for GCN2 using any of these three lines are delayed compared to control Lp-associated larvae. Therefore, we concluded that Lp fails to promote growth when GCN2 expression is altered in enterocytes. We are unsure whether GCN2 is also important for GF larvae’s growth: knock-down using GCN2-1 does not impact pupariation timing in GF, whereas knock-down using GCN2-2 or GCN2-3 does. New Figure 4 shows that GCN2 activity is very low in GF enterocytes. It is possible that despites its low level, GCN2 activity is important for GF larvae to develop, as suggested by the results using lines GCN2-2 and GCN2-3. Lp would then further induce GCN2 expression, improving larval growth through the mechanisms described in Figure 7. On the other hand, the line GCN2-1 is the one generally used in literature (KK line VDRC#103976) and thus seems more reliable. GCN2-2 and GCN2-3 may cause toxicity or off-target effects on GF larvae, which seem to be generally sensitive to RNA interference (see the growth delay of GF larvae upon knock-down of ATF4, 4E-BP or TOR in Figure 6 and Figure 6—figure supplement 1, respectively). Therefore, the importance of GCN2 for the growth of GF larvae is unclear. On the contrary, our data clearly demonstrate the importance of GCN2 in Lp-associated larvae.

What does it look like for these alleles on a balanced diet? Moreover, the day to 50% pupation is variable (eg controls in A different than controls in G). Only one Lp mutant tested and the graphs are spit between H and I while shown altogether in A. Is there a difference between the WT bacteria in the two fly genotypes as seen in A?

We decided to split the H-I panel in two for the sake of clarity, but these data were obtained in the same experiment and thus the data can be grouped. We are including Author response image 3 showing the graph combining Figure 6H and I. The difference between control larvae-WT bacteria and GCN2 knock-down-WT larvae is significant.

**Author response image 3. sa2fig3:** 

Interestingly, AP can rescue independent of GCN2. Thus there are different solutions to this problem. Moreover, the authors found that ATF4 and 4EBP are dispensable, so what is important about GCN2, is it eiF2a phosphorylation. Is translation impacted in enterocytes, is turnover? What is the mechanisms? Does feeding tRNAs impact GCN2 dependent pupation changes? Characterization of the gut cells, ISCs, turnover and physiology (eg metabolomics to monitor aa levels and incorporation)

These are very important questions to address. However, we think that they go beyond the scope of the study, which is centered around the activation of GCN2 by Lp.

…they perform RNAseq and describe a series of GO categories. No experiments either validate or explore the significance of these genes.

To answer this question from the reviewer, we focused on the gene *fezzik*: *fezzik* is a growth suppressor, and we observed that it is down-regulated upon Lp-association in a GCN2-dependant, ATF4-independent and r/tRNA-dependent manner (new Figure 8A). We showed that knock-down of *fezzik* in enterocytes improves larval growth on imbalanced diet (new Figure 8B). We therefore propose that repression of *fezzik* by Lp through r/tRNAs and GCN2 activation participates to improvement of larval growth on imbalanced diet.

We are left with an incomplete understanding of how Lp impacts development, whether RNA is a signal and how this may be sensed to induce a survival program.

As stated in response to the public review, we agree that our understanding of how Lp impacts development and what the signal is remains incomplete. We expect that our results will pave the way to further studies focusing on either signaling through r/tRNAs or on the events downstream GCN2 activation that impact growth. We hope that the reviewer will agree that our new data regarding Valine absorption, extracellular vesicles and *fezzik* help fill the incomplete understanding of how Lp impacts development and how r/tRNAs are sensed.

Experimental interpretations:It is unclear how many independent experiments and how many animals per experiment are included in each figure. What do the aggregate values represent? How many of each? This should be described in detail for all experiments including % pupation experiments and confocal.

We thank the reviewer for this suggestion. We added in the Figure of each legend how many larvae were included in pupation experiments and confocal imaging. We hope it will improve figure’s clarity.

Reviewer #2 (Recommendations for the authors):1) For ease of comprehension, it would be helpful to refer to the L. plantarum mutants by a descriptive designation such as the interrupted tRNA/rRNA or "control", in the case of the control mutant. It is difficult to remember the significance of the number designations.

We thank the reviewer for this suggestion. We replaced “B02.04” with Lp:Tn_control_ and “F07.08” with Lp:Tn_r/tRNA_.

2) Figure 4E and F: Here the authors feed GF *Drosophila* purified bacterial t-RNA's and then show that transcription of 4E-BP is increased in enterocytes. Feeding of an unrelated RNA of similar length would prove that tRNA, in particular, is required for this response.3) Figure S4G: Feeding of both prokaryotic and eukaryotic tRNAs increases expression of 4E-BP. It would be helpful to show an alignment of the sequences of the eukaryotic and prokaryotic tRNAs used here and to comment on their similarity or lack thereof. Is the prokaryotic tRNA similar enough to eukaryotic tRNA to interact with GCN2 if it can be taken up?

In response to points (2) and (3): GCN2 can interact with tRNAs cognate of each amino acid (Masson 2019), which show great sequence dissimilarity. Therefore, it seems more likely that GCN2 recognizes tRNA’s structure rather than tRNA’s sequence.

4) Purified tRNAs have a much smaller effect on 4E-BP expression than L. plantarum. While one possibility is that a bacterial carrier such as OMVs are necessary for uptake into enterocytes, another possibility is that additional bacterial components are required for the response. Does a mixture of heat-killed bacteria and purified tRNAs yield a stronger response? While this experiment would require new reagents, the investigators could prove that transport is necessary by conditionally expressing a prokaryotic tRNA directly in enterocytes and showing that GCN2 is activated.

Ectopic expression of bacterial tRNAs in enterocytes would be of great interest, but we think it goes beyond the scope of our study. As stated above, we now demonstrate that Lp produces extracellular vesicles and that these vesicles contain r/tRNAs (new Figure 3)

5) The data in Figure 5SE showing that L. plantarum does not alter food intake, might be better positioned earlier in the manuscript as this is a key control that does not rely on the data for tRNA or GCN2.

We thank the reviewer for this suggestion. We moved the food intake data to new Figure 1-supplement 1E-F.

6) One of the interesting findings of this paper is that deceleration of development is not the direct result of the inability to access an essential amino acid but rather a regulated process that occurs when nutrition is suboptimal but actually adequate to support faster development. It is interesting to speculate on why this might be an adaptive response.

We share the reviewer’s interest. We now discuss how sensing of symbiotic cues by *Drosophila*’s enterocytes may be an adaptive trait allowing coupling of nutrient absorption and growth with abundance of bacterial symbionts on the substrate (L758-765).

Reviewer #3 (Recommendations for the authors):The work is essentially OK as is, except for a few specific issues outlined above.The comments below aim to improve the impact of the article with suggestions for additional experiments.1. Val absorption: would it be possible to monitor Val intake by the gut in pulse-chase experiments, using labelled valine which most likely would have to be radioactive?

As stated above, we now show that association with Lp improves Valine absorption and Valine levels in hemolymph. Since neither Lp nor the larva can produce Valine de novo, we do not think that it is necessary to use radioactive Valine to demonstrate that the Valine comes from the diet.

2. Release of bacterial r/tRNAs: the authors mention the possibility of export through EVs. A simple experiment is to make an EV preparation and to check by PCR for the presence of r/tRNAs in the purified EVs. This experiment may also allow to answer the question as to whether the bacteria release the r/tRNAs in response to the imbalance of amino-acids in the culture medium (in other words, is the response to the imbalanced diet taking place at the level of the symbiont, the host, or both?). A more challenging one would be to feed the larvae with EVs from wt and mutant Lp bacteria. The difficulty may be to obtain enough material. Does a CAFE-like assay exist for larvae?

We thank the reviewer for this suggestion. As stated above, we did isolate extracellular vesicles from Lp and detected r/tRNAs inside (new Figure 3). We could try to add purified vesicles to the diet, but they would likely be too diluted compared to the concentration of vesicles that could be released inside the gut. A CAFÉ-assay could allow to reach higher concentrations, but to our knowledge such assay is restricted to adults that can “drink” liquid substrates through their proboscis and is not applicable to larvae.

[Editors' note: further revisions were suggested prior to acceptance, as described below.]

The manuscript has been improved but there are some remaining issues that need to be addressed, as outlined below:Please address the specificity questions raised about the vesicles by Reviewer 2.Reviewer #2 (Recommendations for the authors):The authors have responded to all the reviewers' comments in an impressively thorough way. While the effort to observe vesicles in the intestine is admirable, I agree with them that this is not convincing enough to include. I have only one concern regarding the qRT-PCR on membrane vesicles. I would like to see a negative control/ absent RNA that might demonstrate selectivity. The expected RNAs are there. Are these found in the same ratio as in the bacterium? If so, are there any RNAs that are found at different levels in the bacterium and the vesicle? If all RNAs are found in vesicles in the same ratios as in the bacterium, then how to prove that these are not contaminating RNAs? After all, these are some of the most abundant RNAs in the bacterium. I am also concerned that ribosomes may be co-purifying with OMVs. Another possibility might be to treat the membrane vesicles with RNase as RNA in vesicles is thought to be resistant to the action of such enzymes. Of course, I fully appreciate the challenge of proving that OMVs carry rRNAs and tRNAs and the best one may be able to do is provide supporting evidence.

We thank Reviewer 2 for the appreciation of our work and for the suggestions which improved our results. We now provide a comparison of the abundance of 16S rRNA, 23S rRNA and Thr-tRNAs in bacterial cells vs extracellular vesicles. In the new Figure 3, we show that the ratio of the r/tRNAs that we tested strongly differed between cells and extracellular vesicles: in cells, both rRNAs were considerably more abundant than Thr-tRNA but in extracellular vesicles we found them at similar levels (new Figure 3B, C). Moreover, we treated Lp’s extracellular vesicles with RNAse to remove potential contaminants. We then observed a further decrease of rRNAs, which levels got lower than Thr-tRNAs (new Figure 3C, D). These new results strongly suggest that Lp’s extracellular vesicles contain tRNAs, but not-or fewer rRNAs. We modified the Discussion (L964) and the Methods (L1468) to account for these new results.

Reviewer #3 (Recommendations for the authors):The authors have made a reasonable attempt to address the issues raised by the reviewers and it appears that this article has now only to address minor issues. Of note, may-be using RNAScope probes might have given the level of sensitivity required to detect bacterial RNA within enterocytes.

We thank Reviewer 3 for their appreciation of our work and for their suggestions. We will investigate the transfer of bacterial RNA to host cells in further studies.